# Reversible RNA phosphorylation stabilizes tRNA for cellular thermotolerance

Takayuki Ohira[1,4 ✉], Keiichi Minowa[1,4], Kei Sugiyama[1], Seisuke Yamashita[2], Yuriko Sakaguchi[1], Kenjyo Miyauchi[1], Ryo Noguchi[1], Akira Kaneko[3], Izumi Orita[3], Toshiaki Fukui[3], Kozo Tomita[2 ✉] & Tsutomu Suzuki[1 ✉]

Post-transcriptional modifications have critical roles in tRNA stability and function[1–4]. In thermophiles, tRNAs are heavily modified to maintain their thermal stability under extreme growth temperatures[5,6]. Here we identified 2′-phosphouridine (U[p]) at position 47 of tRNAs from thermophilic archaea. U[p]47 confers thermal stability and nuclease resistance to tRNAs. Atomic structures of native archaeal tRNA showed a unique metastable core structure stabilized by U[p]47. The 2′-phosphate of U[p]47 protrudes from the tRNA core and prevents backbone rotation during thermal denaturation. In addition, we identified the *arkI* gene, which encodes an archaeal RNA kinase responsible for U[p]47 formation. Structural studies showed that ArkI has a non-canonical kinase motif surrounded by a positively charged patch for tRNA binding. A knockout strain of *arkI* grew slowly at high temperatures and exhibited a synthetic growth defect when a second tRNA-modifying enzyme was depleted. We also identified an archaeal homologue of KptA as an eraser that efficiently dephosphorylates U[p]47 in vitro and in vivo. Taken together, our findings show that U[p]47 is a reversible RNA modification mediated by ArkI and KptA that fine-tunes the structural rigidity of tRNAs under extreme environmental conditions.

Recent advances in epitranscriptomics research have demonstrated the chemical diversity and biological importance of RNA modifications[1–4]. Thus far, about 150 types of RNA modification have been reported in various RNA molecules from all domains of life[7]. In particular, tRNAs contain the widest variety and largest number of modified nucleosides, with 80% of RNA modifications identified in tRNA molecules. Diverse RNA modifications are clustered in the anticodon loop, especially at positions 34 and 37 (refs. [1,8]). These modifications have critical roles in stabilizing and modulating codon–anticodon interactions on the ribosome, ensuring accurate and efficient protein synthesis. Many RNA modifications are also found in the tRNA body composed of the D-loop, TΨC loop (T-loop) and variable loop (V-loop)[9,10] (Fig. 1a). These RNA modifications are required for correct folding and stability of the tRNA core structure. In particular, 2′-O-methyl modifications (Nm) confer conformational rigidity to the tRNA core region by fixing C3′-endo ribose puckering[9,11].

In thermophilic bacteria and archaea, unique RNA modifications contribute to the thermal adaptation of tRNAs[5,6]. 5-Methyl-2-thiouridine (m[5]s[2]U or s[2]T) is found at position 54 in the T-loop of tRNAs from thermophiles[12]. m[5]s[2]U54 adopts a rigid conformation with C3′-endo ribose puckering, thereby stabilizing the tRNA body in high-temperature environments[11]. The 2-thiolation level of m[5]s[2]U54 increases as the growth temperature rises[13,14]. m[5]s[2]U54 contributes to the thermotolerance of *Thermus thermophilus*[15]. In *Pyrococcus furiosus*, the relative abundance of *N*[4]-acetylcytidine

(ac[4]C) and its 2′-O-methyl derivative (ac[4]Cm) were markedly increased with rising growth temperature[14]. ac[4]C is a prevalent modification that is present in tRNAs, rRNAs and other RNAs in hyperthermophilic archaea[14,16]. ac[4]C favours the C3′-endo form and stabilizes tRNAs[17,18]. Loss of ac[4]C results in a growth defect in *Thermococcus kodakarensis* at high temperature, contributing to cellular thermotolerance[19]. In *Bacillus stearothermophilus*, 2′-O-methylation in tRNAs increases when the growth temperature rises[20]. Archaeosine (G[+]) is a unique 7-deazaguanosine derivative found at position 15 in the D-loop of archaeal tRNAs[21]. On the basis of quantum mechanics calculations, G[+]15 stabilizes the Levitt base pair with C48 (ref. [22]). In line with this, biochemical and genetic studies have shown that G[+] confers thermal stability to tRNAs and contributes to thermotolerance[19,23].

Here we report the identification of 2′-phosphouridine (U[p]) in tRNAs, which, to our knowledge, is the first known instance of internal RNA phosphorylation. Biochemical, structural and genetic studies showed that U[p]47 is a reversible RNA modification and confers thermal stability to tRNA, thereby contributing to cellular thermotolerance.

## Discovery of U[p] in tRNA

To explore tRNA modifications in hyperthermophilic archaea, we isolated 12 tRNA species from the thermoacidophilic crenarchaeon *Sulfurisphaera tokodaii* using our original method for RNA isolation by reciprocal circulating chromatography (RCC) (Extended Data Fig. 1)[24].

[1]Department of Chemistry and Biotechnology, Graduate School of Engineering, The University of Tokyo, Tokyo, Japan. [2]Department of Computational Biology and Medical Sciences, Graduate School of Frontier Sciences, The University of Tokyo, Kashiwa, Japan. [3]School of Life Science and Technology, Tokyo Institute of Technology, Yokohama, Japan. [4]These authors contributed equally: Takayuki Ohira, Keiichi Minowa. ✉e-mail: ohira_t@chembio.t.u-tokyo.ac.jp; kozo-tomita@edu.k.u-tokyo.ac.jp; ts@chembio.t.u-tokyo.ac.jp

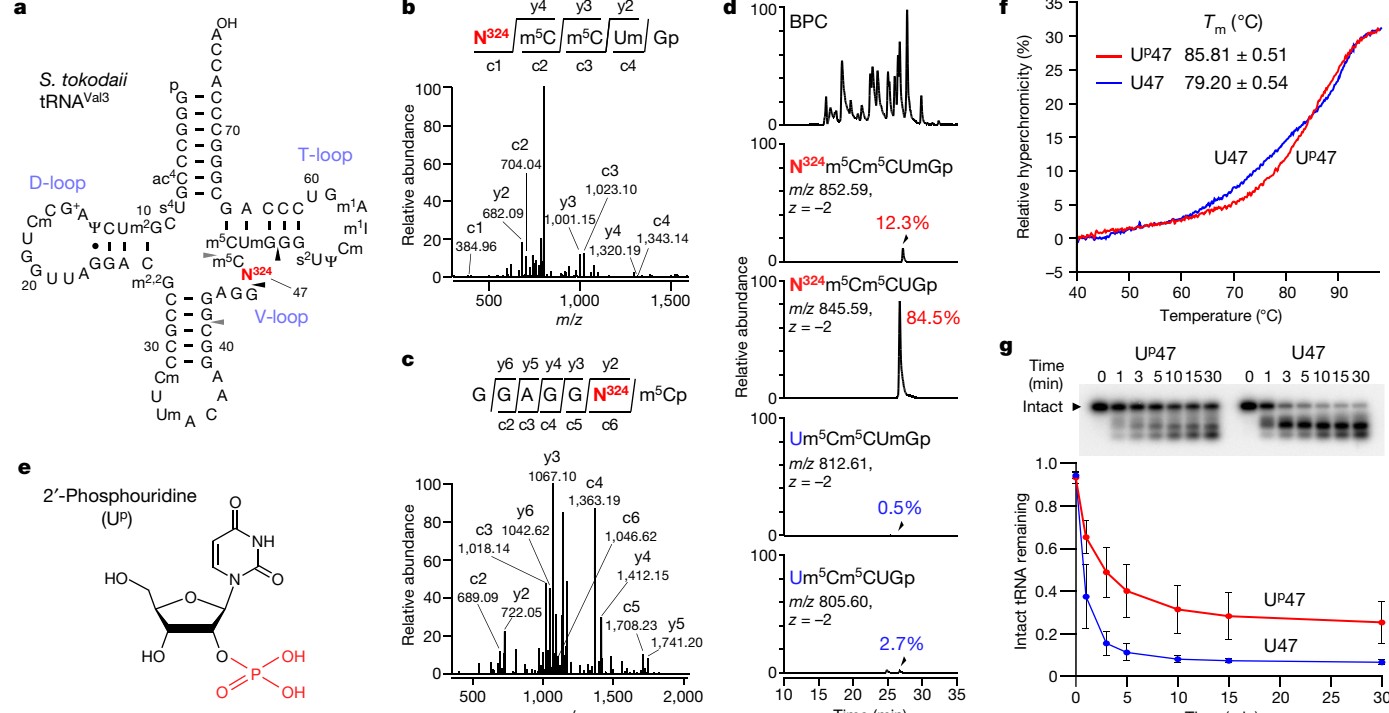

**Fig. 1 | Identification and biochemical characterization of U^p at position 47 of _S. tokodaii_ tRNA^Val3.** **a**, Secondary structure of _S. tokodaii_ tRNA^Val3 with post-transcriptional modifications. $N^{324}$ is shown in red. The cleavage sites of RNase T$_1$ and RNase A that generate RNA fragments with $N^{324}$ are indicated by black and grey triangles, respectively. **b**, CID spectrum of the $N^{324}$-containing fragment ($m/z$ 852.59) of _S. tokodaii_ tRNA^Val2/3 digested with RNase T$_1$. c- and y-series product ions are indicated. **c**, CID spectrum of the $N^{324}$-containing fragment ($m/z$ 1,215.15) of _S. tokodaii_ tRNA^Val2/3 digested with RNase A. c- and y-series product ions are indicated. **d**, RNA-MS of RNA fragments with or without $N^{324}$ and Um50. The upper panel shows the base peak chromatogram (BPC) of RNase T$_1$-digested fragments. The second, third, fourth and fifth panels represent extracted ion chromatograms (XICs) of the divalent negative

ions of the respective fragments, as indicated. **e**, Chemical structure of 2′-phosphouridine (U^p). The 2′-phosphate group is shown in red. **f**, Melting curves of _S. tokodaii_ tRNA^Val3 with (red line) or without (blue line) U^p47. $T_m$ values were determined on the basis of the inflection point of the melting curve. Data represent the average values of technical triplicates ± s.d. $P < 1.02 \times 10^{-4}$ by two-sided Student's $t$ test. **g**, RNase probing of _S. tokodaii_ tRNA^Val3 with (red line) or without (blue line) U^p47. Top, PAGE gels showing degradation of $^{32}$P-labelled tRNA with or without U^p47 by RNase I at the indicated time (min). Intact tRNA is indicated by a triangle. Data represent the average values of technical triplicates ± s.d. The unprocessed gel image is provided in Supplementary Fig. 10.

First, we comprehensively analysed all post-transcriptional modifications of tRNA^Val3 by mass spectrometry (RNA-MS)[25–27] and mapped 13 types of RNA modification at 18 positions in this tRNA molecule (Fig. 1a, Extended Data Fig. 2a–c, Supplementary Note 1, Supplementary Fig. 1, Supplementary Table 1). Among the modifications, we detected an unknown uridine derivative with molecular mass of 324 (tentatively named $N^{324}$) at position 47 of the RNA fragments digested with RNases (Fig. 1b, c). The relative intensity of the mass chromatograms showed that $N^{324}$ occurred at a frequency of 96.8% (Fig. 1d), indicating that $N^{324}$ is an abundant modification. We also detected $N^{324}$ in the seven other tRNA species (Extended Data Fig. 3a, b, Supplementary Note 1, Supplementary Table 2), indicating that $N^{324}$ is a prevalent and abundant (82–100%) modification in class I tRNAs bearing U47 in the V-loop, but not in class II tRNAs with a long V-loop (Extended Data Fig. 3a). We also detected $N^{324}$ in tRNA precursors (Extended Data Fig. 3c, Supplementary Note 1).

High-resolution mass analysis of the $N^{324}$-containing fragment showed that the additional mass of $N^{324}$ attached to the uridine residue was 79.97067 Da, equivalent to one phosphate group (theoretical mass, 79.96632 Da), with a low error value of 4.4 millimass unit, indicating that $N^{324}$ is a phosphorylated uridine residue. This prediction explains why the $N^{324}$ nucleoside was not detected in our nucleoside analysis (Extended Data Fig. 2a), owing to $N^{324}$ being dephosphorylated during nucleoside preparation. To determine the phosphorylation site of $N^{324}$, we prepared the $N^{324}$-containing nucleotide and analysed its chemical

structure through collision-induced dissociation (CID) and biochemical approaches (Supplementary Note 2, Extended Data Fig. 4a–h). We found that phosphorylation occurs on the 2′-OH group of the ribose moiety and concluded that $N^{324}$ is 2′-phosphouridine (denoted U^p, where 'p' is superscript to discriminate it from 3′-phosphate) (Fig. 1e).

## U^p47 stabilizes tRNA structure

Given that U^p47 is a thermophile-specific modification found in the tRNA core region, we investigated whether U^p47 stabilizes the tertiary structure of tRNA. To this end, we treated _S. tokodaii_ tRNA^Val3 with yeast Tpt1p (2′-phosphotransferase) to remove the 2′-phosphate of U^p47. We measured the melting temperature ($T_m$) of _S. tokodaii_ tRNA^Val3 with and without U^p47 (Fig. 1f). In the melting curves, the tRNA without U^p47 gradually melted at around 65 °C while its hyperchromicity increased with temperature, whereas the tRNA with U^p47 remained stable even at 70 °C. The $T_m$ values of the tRNA with and without U^p47 were 85.8 ± 0.5 °C and 79.2 ± 0.5 °C, respectively. These observations clearly demonstrate that a single U^p47 modification increases the thermal stability of tRNA^Val3 by 6.6 °C.

We next performed an RNase probing experiment to examine the nuclease resistance of tRNA with and without U^p47. _S. tokodaii_ tRNA^Val3 and its Tpt1p-treated form were labelled with $^{32}$P at their 3′ termini and were probed with RNase I at 65 °C (Fig. 1g). Over time, the intact tRNAs were gradually degraded into RNA fragments. Compared with

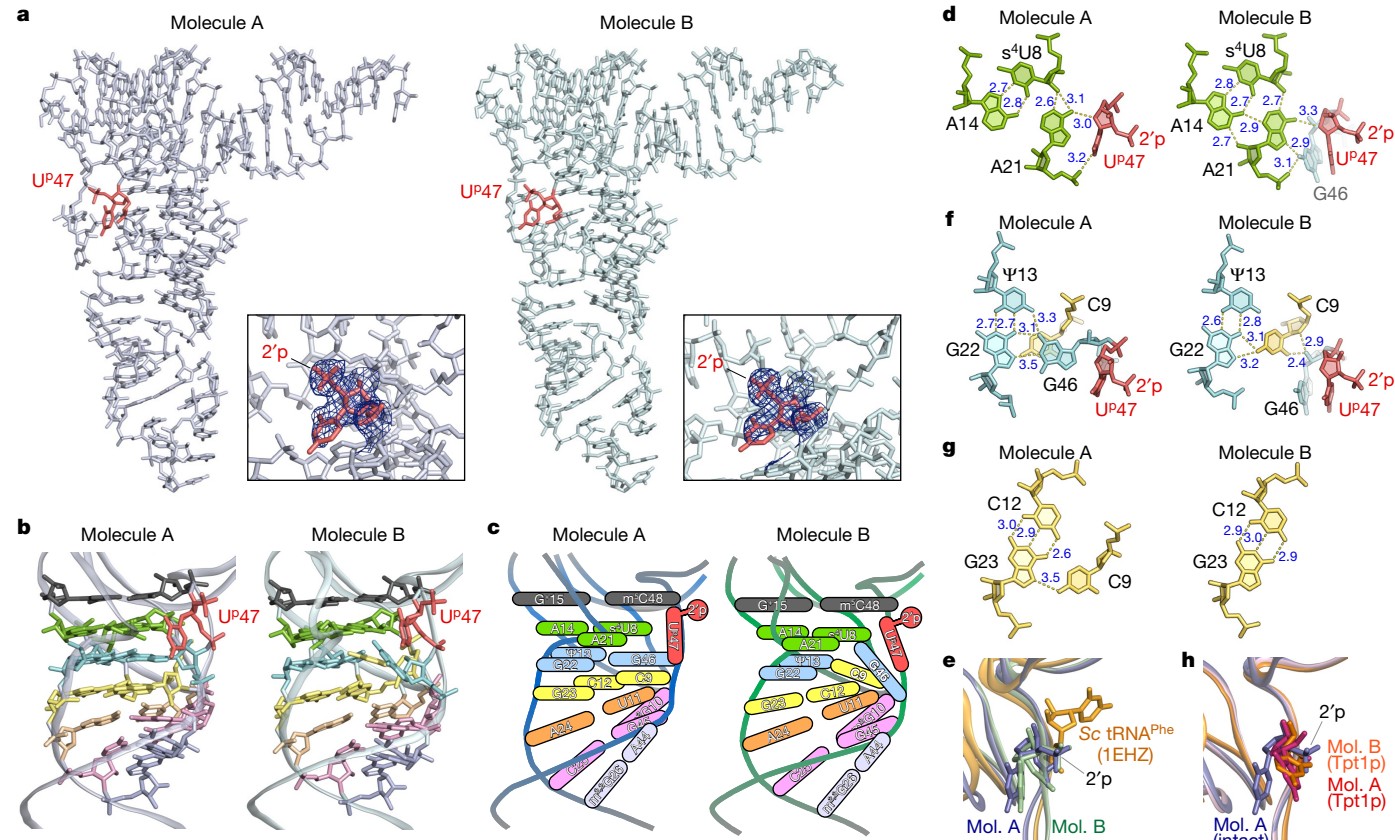

**Fig. 2 | Structural characterization of U$^p$47 in *S. tokodaii* tRNA$^{Val3}$.**
**a**, Overview of the crystal structures of *S. tokodaii* tRNA$^{Val3}$. Molecules A and B are shown in stick representation in white and light green, respectively. U$^p$47 is shown in red. The $2F_o$–$F_c$ electron density map contoured at 1.0σ around U$^p$47 is shown in the lower right box for each molecule. 2′p, 2′-phosphate. **b**, Close-up views of the core structures of molecules A (left) and B (right). U$^p$47 (red) and the top (15–48; black), second (8–14–21; green), third (13–22–46; blue), fourth (9–12–23; yellow), fifth (11–24; orange), sixth (10–25–45; pink) and seventh (26–44; light purple) layers are shown in stick representation. **c**, Schematic views of the core structures of molecules A (left) and B (right). **d**, **f**, **g**, Atomic

structures of the base triples s$^4$U8–A14–A21 (**d**), Ψ13–G22–G46 (**f**) and C12–G23–C9 (**g**) in the core region of molecules A (left) and B (right). Dashed lines indicate predicted interactions with bond length (Å). U$^p$47 is shown in red. **e**, The V-loop structures of molecule (mol.) A (blue), molecule B (green) and *Saccharomyces cerevisiae* tRNA$^{Phe}$ (PDB, 1EHZ) (gold) are overlaid. The residues at position 47 are shown in stick representation. The 2′-phosphate of U$^p$47 is indicated. **h**, The V-loop structures of intact tRNA molecule A (blue), Tpt1p-treated tRNA molecule A (magenta) and Tpt1p-treated tRNA molecule B (orange) are overlaid. The residues at position 47 are shown in stick representation. The 2′-phosphate of U$^p$47 is indicated.

the intact tRNA with U$^p$47, the Tpt1p-treated tRNA was degraded more rapidly, within 5 min, indicating that the tRNA without U$^p$47 was highly sensitive to RNase I. This observation demonstrates that U$^p$47 stabilizes and protects tRNAs from nucleolytic degradation.

## Structural study of U$^p$47 in native tRNA

To determine the molecular basis for thermal stabilization of tRNA by U$^p$47, we crystalized *S. tokodaii* tRNA$^{Val3}$ and determined its atomic structure at a resolution of up to 1.9 Å by X-ray crystallography (Fig. 2a, Extended Data Table 1, Extended Data Fig. 5, Supplementary Note 3, Supplementary Fig. 2). One unit cell contains two tRNA molecules, denoted as molecule A and molecule B. Molecule A formed a canonical tRNA core structure (Fig. 2b, c, Extended Data Fig. 5a), whereas molecule B had an altered core structure with a non-canonical base triple (Fig. 2b, c, Extended Data Fig. 5b). We clearly observed electron densities for tRNA modifications, including U$^p$47 (Fig. 2a, Extended Data Fig. 5c, d). In both molecules, the 2′-phosphate of U$^p$47 was oriented towards the solvent side and did not interact with any residues (Fig. 2a–c). The ribose puckering of U$^p$47 adopted a C2′-*endo* conformation (Supplementary Table 3), as observed in the synthetic nucleotide[28]. The O4′ position in the ribose of U$^p$47 formed a hydrogen bond with the $N^6$-amino group of A21 in both molecules (Fig. 2d). The uracil

base of U$^p$47 faced the tRNA core (Fig. 2b, d). Because the uracil base at position 47 favours an outward orientation, as observed in well-known structures of yeast tRNA$^{Phe}$ and other class I tRNAs[29], we observed backbone rotation of the V-loop at positions 46–48 caused by U$^p$47 (Fig. 2e, Extended Data Fig. 6a, b). When U$^p$47 was virtually introduced to yeast tRNA$^{Phe}$, the 2′-phosphate clashed with T-stem residues at positions 49 and 50, inducing backbone rotation of the V-loop that orients the uracil base inwards and the 2′-phosphate outwards (Fig. 2e). In this rotation from yeast tRNA$^{Phe}$ to molecule A (Fig. 2b, c, e, Extended Data Fig. 6b), G46 changed its ribose pucker from C2′-*endo* to C3′-*endo* with altered torsion angles (δ, ε and ζ were changed by −58°, −28° and 68°, respectively) (Extended Data Fig. 6b, Supplementary Table 3). In addition, the U$^p$47 backbone was substantially rotated with the α and ζ angles changing by −113° and 167°, respectively (Extended Data Fig. 6b, Supplementary Table 3). The m$^5$C48 backbone was also rotated, with the α and γ angles changing by −36° and −86°, respectively (Extended Data Fig. 6b, Supplementary Table 3).

Although molecule A had a canonical tRNA core structure stabilized by multiple tertiary interactions between the D-arm and V-loop, including the base triples s$^4$U8–A14–A21, Ψ13–G22–G46, C12–G23–C9 and m$^2$G10–C25–G45 (Fig. 2b, c), molecule B unexpectedly had a non-canonical core structure (Fig. 2b, c). In molecule B, G46 was dissociated from the base triple Ψ13–G22–G46 and stacked with U$^p$47

(Fig. 2f, Supplementary Video 1). The $N^2$-amino group of G46 formed hydrogen bonds with A21 by inserting itself between the base triple and $U^P47$ (Fig. 2d). This interaction pushes A21 towards A14 to make additional hydrogen bonds that stabilize the s$^4$U8–A14–A21 triple (Fig. 2d). Because $U^P47$ does not substantially change its backbone conformation (Extended Data Fig. 6b, Supplementary Table 3, Supplementary Video 1), the G46 base was stably trapped by $U^P47$ in molecule B (Fig. 2d, f). To compensate for this conformational change, C9 comes up from the lower layer (C12–G23–C9) (Fig. 2g, Supplementary Video 1) to form the non-canonical base triple Ψ13–G22–C9 (Fig. 2f, Supplementary Video 1). Thus, the molecule B structure has a non-canonical base triple that might be stabilized by $U^P47$. In this structural alteration, the torsion angles of A44 and G45 were slightly changed to make the backbone bulge outwards, flipping the G46 base out with the χ angle altered by −70° (Extended Data Fig. 6b, Supplementary Table 3). C9 changes its backbone, altering the α, β, γ and χ angles by 171°, −37°, −180° and 26°, respectively (Supplementary Table 3).

To further investigate the structural role of $U^P47$, we also solved a crystal structure for Tpt1p-treated *S. tokodaii* tRNA$^{Val3}$ (Extended Data Fig. 7a). Both molecules A and B of the Tpt1p-treated tRNA showed the canonical structure with the standard core (Extended Data Fig. 7b–f). In both molecules, U47 was dissociated from the s$^4$U8–A14–A21 base triple (Extended Data Fig. 7b–e) with backbone angles α, γ and ε altered by 153°, −109° and −37°, respectively (molecule A) (Extended Data Fig. 6b, Supplementary Table 3), thereby placing the uracil base of U47 outwards (Fig. 2h, Extended Data Fig. 7b–d). In another aspect of the Tpt1p-treated tRNA, C9 was detached from the C12–G23–C9 base triple in both molecules (Extended Data Fig. 7f). These findings imply that $U^P47$ stabilizes the metastable tRNA core structure with a non-canonical base triple during thermal denaturation.

## Identification of an RNA kinase for $U^P47$

To identify a gene responsible for $U^P47$ formation, we narrowed down the candidate genes in the *S. tokodaii* genome by performing a comparative genomic analysis of sequenced genomes using RECOG (http://mbgd.genome.ad.jp/RECOG/). According to our analysis of $U^P47$ distribution in archaeal species (Supplementary Note 4, Extended Data Fig. 8a–d), $U^P47$ is present in seven archaeal species, including in *S. tokodaii*, but is absent in two species (Fig. 3a). Among the 2,826 genes encoded in the *S. tokodaii* genome, only nine genes (Supplementary Table 4) were commonly found in all seven archaeal species with $U^P47$ (Fig. 3b). Among them, five genes (Supplementary Table 4) were of uncharacterized function (Fig. 3b). We chose one gene encoding a putative protein kinase, STK_09530 (hypothetical serine/threonine kinase, COG2112), as a strong candidate (Fig. 3b). STK_09530 resides in an operon containing a gene for a tRNA nucleotidyltransferase (STK_09520), implying that it encodes an enzyme related to tRNA maturation. We then constructed a strain of *T. kodakarensis* lacking *tk2051*, an orthologue of STK_09530. The tRNA fraction obtained from the Δ*tk2051* strain was subjected to liquid chromatography followed by MS (LC–MS) nucleotide analysis. A pU$^P$m$^5$C dimer was clearly observed in the parental strain (wild type) of *T. kodakarensis* (KU216), but was absent in the Δ*tk2051* strain (Fig. 3c). Therefore, *tk2051* is the gene responsible for $U^P47$ formation in cells. We designated the gene *arkI* (archaeal RNA kinase).

## $U^P47$ confers cellular thermotolerance

Next, we investigated the physiological importance of $U^P47$ in *T. kodakarensis*. The Δ*arkI* strain grew as well as the wild-type strain (KU216) at the nearly optimal temperature of 83 °C (Fig. 3d), whereas it showed a weak temperature-sensitive phenotype with slower growth than the wild-type strain at 87 °C and 91 °C (Fig. 3d). We considered synthetic effects of $U^P47$ with other tRNA modification, thus constructing a Δ*arkI*Δ*queE*

double-knockout strain, in which *queE* is responsible for archaeosine (G$^+$15) formation, because G$^+$15 thermally stabilizes tRNAs and contributes to cellular thermotolerance[19]. We confirmed the absence of $U^P47$ and G$^+$15 in tRNAs from the double-knockout strain (Supplementary Fig. 3). The Δ*queE* strain grew well at 83 °C, slowly at 87 °C and not at all at 91 °C (Fig. 3d), as reported[19]. The Δ*arkI*Δ*queE* strain grew slower than the wild-type, Δ*arkI* and Δ*queE* strains at 83 °C (Fig. 3d). The strain exhibited a severe growth phenotype at 87 °C (Fig. 3d) and was unable to survive at 91 °C (Fig. 3d). This finding indicates that $U^P47$ and G$^+$15 cooperatively stabilize the tRNA core structure at high temperatures, thereby contributing to cellular thermotolerance.

## Kinetics of tRNA phosphorylation by ArkI

We prepared recombinant *T. kodakarensis* ArkI (TkArkI) and examined in vitro $U^P47$ formation. $U^P47$ was efficiently reconstituted only in the presence of ATP (Fig. 3e). We then performed kinetic measurement of $U^P47$ formation catalysed by TkArkI. The $K_m$ and $V_{max}$ values for tRNA were 97.3 nM and 9.9 nM min$^{-1}$, respectively (Fig. 3f), showing that TkArkI efficiently recognizes tRNA substrate. By contrast, the $K_m$ value for ATP was found to be 1.2 mM (Fig. 3f). This value is extremely high when compared with the values for known protein kinases. This finding indicates that TkArkI-mediated $U^P47$ formation might be regulated by sensing the cellular ATP concentration. We also characterized ArkI homologues from other archaeal and bacterial species (Supplementary Note 5, Supplementary Figs. 4, 5a–c).

## Crystal structure of TkArkI

To find the structural basis of $U^P47$ formation, we crystallized TkArkI and determined its atomic structure at a resolution of 1.8 Å using X-ray crystallography (Fig. 4a, Extended Data Table 1). On the basis of its amino acid sequence, TkArkI belongs to a superfamily of eukaryotic protein kinases (ePKs)[30]. As observed for ePKs, TkArkI also consisted of two lobes, termed the N-terminal and C-terminal lobes, which were connected by a hinge (positions 96–109) (Fig. 4a, Extended Data Fig. 9). ePKs consist of 12 conserved subdomains that fold into the catalytic core. TkArkI had subdomains I–V in the N-terminal lobe and subdomains VIab, VII, IX and XI in the C-terminal lobe, but lacked subdomains VIII and X (Fig. 4b, Extended Data Fig. 9). The conserved motifs of the P-loop (positions 31–38), catalytic loop (positions 128–140) and metal-binding loop (positions 145–153) were present in subdomains I, VIb and VII, respectively (Fig. 4b, Extended Data Fig. 9). Compared with the canonical ePK, the characteristic sequences in the conserved motifs were altered in TkArkI. The HRD triplet in the catalytic loop (VIb) of ePKs was replaced with HGQ in TkArkI (Extended Data Fig. 9). In addition, the DFG triplet in the metal-binding loop (VII) of ePKs was replaced with DFE in TkArkI (Extended Data Fig. 9). In subdomain IX, TkArkI had an α-helix (α6) specific to ArkI homologues. In subdomain XI, TkArkI had a longer α-helix (α7), when compared with the same helix in mouse PRKACA. In the extended C terminus of α7, the YKR motif is conserved in ArkI-family proteins (Extended Data Fig. 9), indicating that this positively charged motif is involved in RNA binding.

Although we demonstrated that TkArkI is an ATP-dependent RNA kinase involved in the formation of $U^P47$ (Fig. 3e, f), we observed a clear electron density for guanosine in the cleft of the two lobes (Fig. 4a, c, d), which corresponds to the ATP-binding site of ePKs surrounded by the hinge and metal-binding, catalytic and P-loops (Supplementary Fig. 6a, b). We confirmed guanosine (and deoxyguanosine) as a ligand that tightly binds to TkArkI (Supplementary Note 6, Supplementary Fig. 7a–c). These observations indicate that TkArkI has binding affinity for guanosine and deoxyguanosine but uses ATP as a major phosphate donor. In the ATP-binding site of mouse PRKACA (Supplementary Fig. 6a, b), the triphosphate of ATP coordinates two

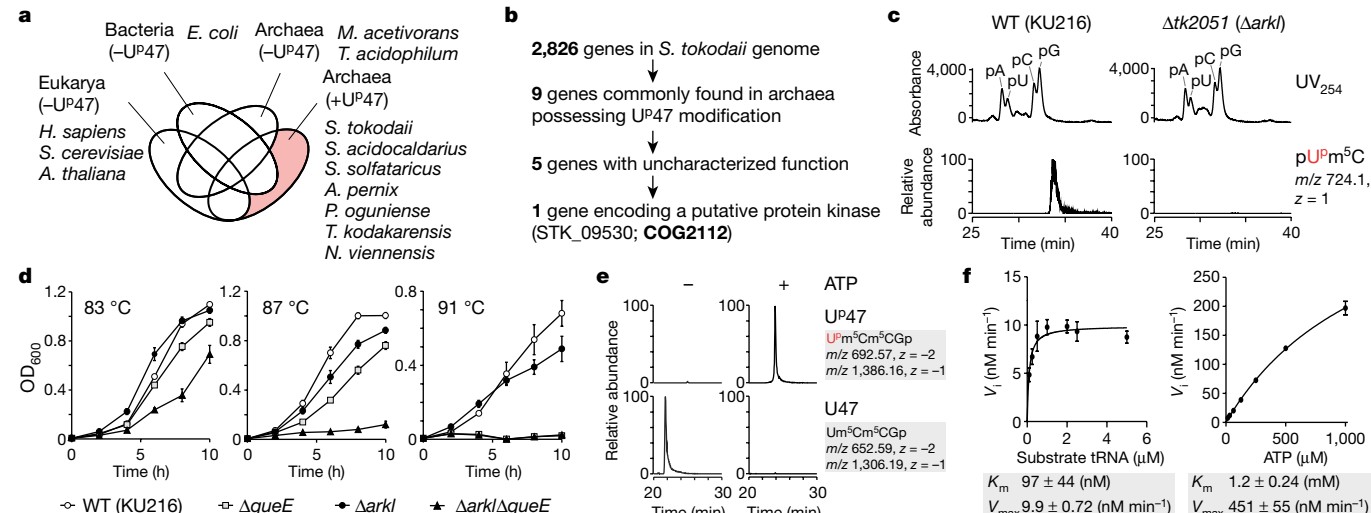

**Fig. 3 | Identification and characterization of the RNA kinase responsible for Uᵖ47 and its physiological role. a**, Venn diagram showing unique and shared genes among the Bacteria (*E. coli*), Eukarya (*Homo sapiens*, *S. cerevisiae* and *Arabidopsis thaliana*) and Archaea (*Methanosarcina acetivorans*, *Thermoplasma acidophilum*, *S. tokodaii*, *Sulfolobus acidocaldarius*, *Saccharolobus solfataricus*, *Aeropyrum pernix*, *Pyrobaculum oguniense*, *T. kodakarensis* and *Nitrososphaera viennensis*) domains possessing (+) or lacking (−) Uᵖ47. The pale red area includes genes unique to archaea having Uᵖ47. **b**, Comparative genomic analysis performed to narrow down the candidate genes responsible for Uᵖ47 modification. **c**, LC–MS nucleotide analysis of tRNA fractions from wild-type (WT, KU216) (left) and Δ*tk2051* (right) strains of *T. kodakarensis*. The upper panel shows the UV trace at 254 nm. The peaks for pA, pU, pC and pG are marked. The lower panel shows the XIC for the proton adduct of the dimer pUᵖm⁵C (*m/z* 724.1, *z* = 1). **d**, Growth measurement (OD₆₀₀) of wild-type (KU216) (open circles), Δ*queE* (squares), Δ*arkI* (closed circles) and Δ*arkI*Δ*queE* (triangles) strains of *T. kodakarensis* at 83 °C (left), 87 °C (middle) and 91 °C (right). Data represent the average values of technical triplicates ± s.d. **e**, In vitro reconstitution of Uᵖ47 with recombinant TkArkI in the presence (right panels) or absence (left panels) of ATP. XICs show the sum of monovalent and divalent negative ions from RNase T₁-digested fragments containing Uᵖ47 (upper panels) or U47 (lower panels). **f**, Kinetic measurements of in vitro Uᵖ47 formation by TkArkI. The initial velocity ($V_i$) of the phosphorylation reaction was measured at the indicated concentrations of tRNA (left) and ATP (right). Data represent the average values of technical triplicates ± s.d. The $K_m$ and $V_{max}$ values are shown below each graph.

Mn²⁺ ions and interacts tightly with the conserved motifs, especially the metal-binding loop and P-loop. However, in the guanosine-bound TkArkI structure, the P-loop was dislocated from the ligand-binding site (Fig. 4c, d). Thus, ATP does not bind the ligand-binding site of the observed structure. In homology modelling to ePKs (Supplementary Fig. 6c), ATP virtually bound to the active form of the ligand-binding site of TkArkI. It is likely that the P-loop and other motifs form the active pocket for ATP binding following tRNA binding to TkArkI. Although the biological relevance of guanosine binding to TkArkI is not known, guanosine may compete with ATP to regulate tRNA phosphorylation, similar to the mechanism by which nucleoside derivatives inhibit protein kinases[31,32]. Judging by its high $K_m$ value for ATP (1.2 mM) (Fig. 3f), TkArkI might sense the cellular energy status and guanosine binding to TkArkI might have a regulatory role in Uᵖ47 formation. Given that TkArkI was a recombinant protein expressed in *Escherichia coli*, we cannot rule out the possibility that guanosine was an artificial ligand bound to the inactive form of TkArkI. It is unclear whether guanosine actually binds to TkArkI within archaeal cells at high growth temperatures.

The electrostatic surface potential showed a large positive area on one side of the TkArkI structure (Fig. 4e). The positively charged surface covered the ATP-binding site in the N-terminal lobe and extended to the ArkI-specific elongated α7 helix in the C-terminal lobe (Extended Data Fig. 9). Instead of the missing subdomain VIII involved in recognition of substrate peptide in ePKs (mouse PRKACA), the basic surface in the C-terminal lobe might bind substrate tRNA through electrostatic interaction.

To characterize the conserved residues in TkArkI, we constructed 14 TkArkI mutants in which targeted residues were replaced by alanine (Fig. 4b, f). All mutants were expressed in soluble form and purified. The tRNA phosphorylation activity of each mutant was measured (Fig. 4g). In the ATP-binding site, K32A, G33A, K51A and E65A substitutions markedly reduced activity, whereas the R95A substitution caused a mild reduction in activity. In addition, a severe reduction in activity was observed in the H130A, Q132A and K137A mutants with substitutions in the catalytic loop. No activity was detected for the D149A mutant, in which the mutated residue is in subdomain VII involved in metal binding. These results clearly confirm the critical role of catalytic residues in kinase activity. The N160A and T162A substitutions in subdomain IX led to decreased activity. We mutated the YKR motif in the α7 helix, finding a severe reduction in activity with the K201A substitution and a mild reduction with the Y200A and R202A substitutions. These observations indicate the importance of the conserved residues and positively charged surface in the C-terminal lobe.

## KptA acts as an eraser for Uᵖ47

Tpt1p removes the 2′-phosphate from tRNA precursors during maturation[33]. Tpt1/KptA homologues are distributed across all domains of life[34,35] (Supplementary Fig. 4). Although Tpt1/KptA homologues are also present in thermophilic archaea and bacteria (Supplementary Fig. 4), natural RNA substrates with 2′-phosphate have not been identified.

Efficient removal of Uᵖ47 by yeast Tpt1p prompted us to speculate that archaeal KptA is capable of removing the 2′-phosphate of Uᵖ47 from tRNAs in the cell (Fig. 5a). To explore this possibility, we conducted in vitro dephosphorylation of Uᵖ47 with *T. kodakarensis* KptA (TkKptA) in the presence of NAD⁺, with the results indicating that the 2′-phosphate of Uᵖ47 was efficiently removed (Fig. 5b). In the same reaction conditions used for Uᵖ47 formation by TkArkI, we measured the kinetic parameters of Uᵖ47 dephosphorylation catalysed by TkKptA: the $K_m$ and $V_{max}$ values for tRNA were 180 nM and 27 nM s⁻¹, respectively (Fig. 5c). The $K_m$ value for dephosphorylation by TkKptA is comparable to that of phosphorylation by TkArkI, implying that TkKptA acts as an eraser for Uᵖ47 in the cell.

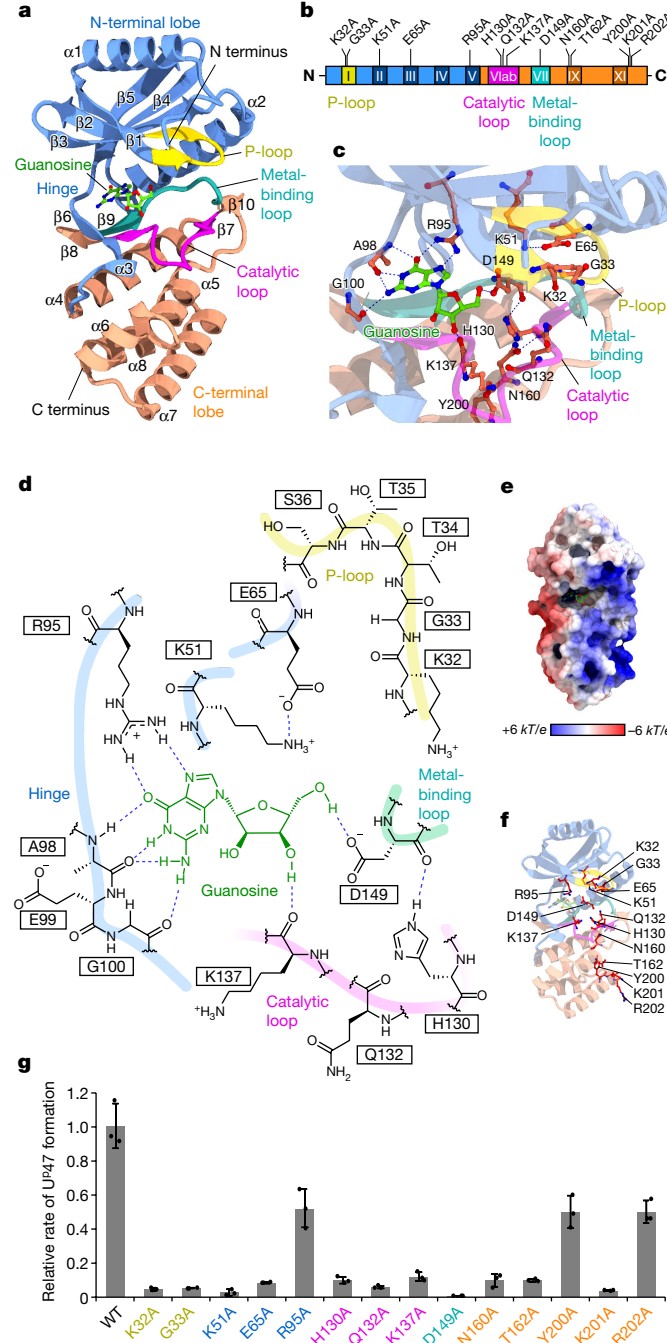

**Fig. 4 | Crystal structure and characterization of TkArkI. a**, Overall structure of TkArkI with five features highlighted: the N-terminal lobe (residues 1–30 and 39–109; blue), P-loop (residues 31–38; yellow), C-terminal lobe (residues 110–127, 141–144 and 154–216; orange), subdomain VIb (catalytic loop, residues 128–140; pink) and subdomain VII (metal-binding loop, residues 145–153; cyan). Guanosine observed in a putative ATP-binding pocket is shown in ball-and-stick representation. **b**, Subdomains of TkArkI showing the locations of mutations examined in this study. Colour codes for each feature are the same as in **a**. **c**, Close-up view of the putative ATP-binding pocket in TkArkI. Residues for which mutations were examined in this study are indicated. Guanosine is shown in ball-and-stick representation. **d**, Schematic diagram of guanosine binding in the putative ATP-binding pocket. Predicted interactions are indicated with dashed lines. The main chains of the P-loop, hinge, catalytic loop and metal-binding loop are shown with bold lines. **e**, Electrostatic surface potential of TkArkI. Positively and negatively charged areas are coloured in blue and red, respectively. Guanosine is shown in ball-and-stick representation. The surface potential is described as dimensionless numbers. k*T*/e refers to the conversion factor (k, proportion constant; *T*, temperature; e; charge unit). **f**, Positions of mutation sites indicated in the crystal structure. **g**, Relative activities of a series of TkArkI mutants, normalized against the activity of wild-type TkArkI. Data represent the average values of technical triplicates ± s.d.

when TkKptA expression was induced by addition of 10 or 100 μM IPTG (Fig. 5d, e). All four U^p47-containing fragments had decreased abundance as a function of IPTG concentration, demonstrating that TkKptA erases U^p47 in *E. coli*. We obtained similar results with *E. coli* KptA (Extended Data Fig. 10a, b) and *S. cerevisiae* Tpt1p (Extended Data Fig. 10c, d). Together, these data demonstrate that Tpt1/KptA homologues dephosphorylate U^p47 of tRNAs in vivo.

## Discussion

U^p47 is, to our knowledge, the first known instance of internal phosphorylation as a stable RNA modification (Supplementary Note 7). 2′-Phosphate at an internal residue appears transiently during tRNA splicing in fungi and plants[36,37]. However, this moiety is not formed by phosphorylation but rather through hydrolysis of 2′,3′-cyclic phosphate via the healing and sealing pathway[36,38]. Because the 2′-phosphate is removed by Tpt1p[33], it is not present in mature tRNAs.

In *S. tokodaii* tRNAs isolated in this study, U^p47 was detected in nine class I tRNA species with high frequency (82–100%) (Fig. 1d, Extended Data Fig. 3a) but was absent in two class I tRNAs (tRNA^Gln2 and tRNA^Cys) and two class II tRNAs (tRNA^Leu4 and tRNA^Ser3) (Extended Data Fig. 3a). Judging from the primary sequences of these species (Supplementary Fig. 9), it is likely that ArkI introduces U^p47 in tRNAs bearing a V-loop with five bases, as tRNA^Gln2 and tRNA^Cys have four and six bases in the V-loop, respectively. Supporting this finding, only the class I tRNA fraction was phosphorylated in total RNA by in vitro reaction (Supplementary Fig. 5b, c).

RNA hydrolysis is mediated by the 2′-OH group in the presence of divalent metal ions such as Mg^{2+}. Especially at high temperatures, RNA is rapidly degraded. Similarly to 2′-O-methylation, the 2′-phosphorylation of U^p47 also serves to prevent tRNA degradation. This property partly explains the RNase resistance of tRNA conferred by U^p47 (Fig. 1g). It is known that U^p adopts C2′-*endo* ribose puckering[28], which confers flexibility to the RNA strand by extending the backbone structure[39]. Hence, U^p47 presumably acts as a defining mark for single-stranded RNA. In the process of tRNA folding, U^p47 might have a role in preventing the V-loop from being accidentally incorporated into stem structures, ensuring correct folding of the tRNA L-shape structure. Especially in thermophiles, tRNA might frequently misfold owing to its high G+C content. Thus, U^p47 deposition in the tRNA precursor might be required to loop out the V-loop region to ensure correct folding of the tRNA. Other modifications at position 47, acp^3U[40] and dihydrouridine[7], are used in bacteria and eukaryotes, respectively. acp^3U directly prevents

We then examined the in vivo function of Tpt1/KptA homologues in U^p47 dephosphorylation, using *E. coli* as a model organism. Because *E. coli* tRNAs have m^7G46 and acp^3U47 modifications, which inhibit U^p47 formation in the V-loop, we used the *E. coli* Δ*trmB*Δ*tapT* strain as a host cell in which both of these tRNA modifications are absent and then expressed *Nitrososphaera viennensis* ArkI (NvArkI), because *N. viennensis* is a mesophilic archaeon and its ArkI homologue was predicted to have efficient activity in *E. coli*. The class I tRNA fraction prepared from this strain was subjected to shotgun analysis to detect the U^p47 modification. We clearly detected four U^p47-containing fragments derived from various *E. coli* tRNA species (Fig. 5d, Supplementary Table 5). Each fragment was sequenced by higher-energy collision dissociation analysis, confirming the presence of U^p at position 47 (Supplementary Fig. 8). Next, we introduced TkKptA under the control of an isopropyl β-ᴅ-1-thiogalactopyranoside (IPTG)-inducible promotor and quantified the peak intensity of each U^p47-containing fragment

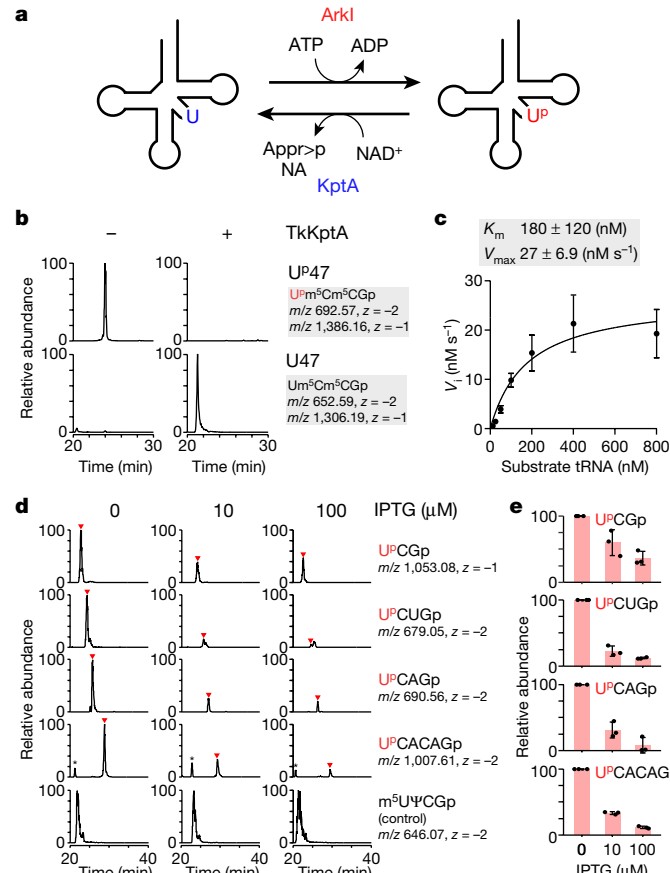

**Fig. 5 | KptA acts as a potential eraser for $U^P$47. a**, Reversibility of $U^P$47 mediated by ArkI and KptA. ArkI phosphorylates U47 of tRNA to form $U^P$47 using ATP as a phosphate donor, producing ADP as a by-product. KptA converts $U^P$47 to U47 by transferring the phosphate group of $U^P$47 to $NAD^+$, producing ADP-ribose 1′′,2′′-cyclic phosphate (Appr>p) and nicotinamide (NA) as by-products. **b**, In vitro dephosphorylation of $U^P$47 with (+) or without (−) TkKptA. XICs show the sum of monovalent and divalent negative ions from RNase $T_1$-digested fragments containing $U^P$47 (upper panels) or U47 (lower panels). **c**, Kinetic measurement of in vitro $U^P$47 dephosphorylation by TkKptA. The initial velocity ($V_i$) of the dephosphorylation reaction was measured at the indicated tRNA concentrations. Data represent the average values of technical triplicates ± s.d. The $K_m$ and $V_{max}$ values are shown above the graph. **d**, Dephosphorylation of $U^P$47 by TkKptA in *E. coli*. XICs show $U^P$47-containing fragments derived from various *E. coli* tRNA species (Supplementary Table 5) from an *E. coli* $\Delta trmB\Delta tapT$ strain expressing *N. viennensis* ArkI: $U^P$CGp (top panels), $U^P$CUGp (second panels), $U^P$CAGp (third panels), $U^P$CACAGp (fourth panels) and $m^5U\Psi$CGp as a control fragment (bottom panels). Relative abundance of the $U^P$47-containing fragments was measured in *E. coli* strains in which TkKptA was not expressed (left panels) or where TkKptA expression was induced with 10 μM (middle panels) or 100 μM (right panels) IPTG. **e**, Relative peak intensity of each $U^P$47-containing fragment detected in the tRNA fraction from *E. coli* strains cultured with 0, 10 or 100 μM IPTG. Data represent the average values of technical triplicates ± s.d.

the V-loop from being incorporated into stem structures by inhibiting base pairing. Dihydrouridine also adopts the C2′-*endo* conformation[41] and confers flexibility to the V-loop. It is interesting that similar functions are evolutionarily conserved in different V-loop modifications across the domains of life.

Intriguingly, *S. tokodaii* tRNA$^{Val3}$ was present as two isomers (molecules A and B) with different conformations in the core region (Fig. 2a). Molecule A has a standard core structure found in many tRNAs, whereas molecule B has a non-standard core structure. Because the Tpt1p-treated tRNA has the canonical structure with the standard

core (Extended Data Fig. 7a–f), it is likely that the structural alteration is caused by $U^P$47. During thermal denaturation of tRNAs, the core region and D-arm are unwound first[42,43]. In molecule B, G46 is released from the base triple $\Psi$13–G22–G46 and stacks with the uracil base of $U^P$47 (Fig. 2d, f). Presumably, this unique conformation is a metastable structure of tRNA during heat denaturation. Curiously, in the structural transition from molecule A to molecule B (Supplementary Video 1), the torsion angle of G46 changes substantially, whereas that of $U^P$47 does not (Extended Data Fig. 6b). $U^P$47 catches the G46 base that is dissociated from the base triple to restrict further rotation of the V-loop, thereby stabilizing the metastable core structure of the tRNA to prevent its heat denaturation. In addition, C9 comes up from the lower layer (C12–G23–C9) to fill in for the missing G46, forming the non-canonical base triple $\Psi$13–G22–C9 (Fig. 2f). $U^P$47 does not fix the tRNA rigidly but rather maintains a metastable structure when the tRNA core thermally fluctuates, thereby preventing further collapse of the core structure, as well as increasing the chance of return to the canonical structure.

ArkI homologues are mainly distributed in thermophilic archaea but are also present in some bacteria (Supplementary Fig. 4). We confirmed the activity of tRNA phosphorylation for bacterial ArkI homologues (Supplementary Fig. 5a, c). In silico analysis of protein kinases suggested that ArkI-family proteins were originally classified as members of the AQ578 family found in bacterial and archaeal genomes[44]; the AQ578 family was proposed to have emerged by gene duplication in the early archaeal lineage. The bacterial AQ578 family might have been acquired by horizontal gene transfer of the archaeal homologue, suggesting that the strategy of stabilizing tRNA by internal phosphorylation might have spread across the domains of life.

The $\Delta arkI$ strain of *T. kodakarensis* exhibited weak temperature sensitivity (Fig. 3d), demonstrating that $U^P$47 by itself contributes to cellular thermotolerance. Because multiple tRNA modifications cooperatively stabilize the tRNA structure, we chose to analyse the $G^+$15 modification, showing a synthetic phenotype with $U^P$47 loss. We found that the $\Delta arkI\Delta queE$ double-knockout strain was extremely susceptible to high temperature (Fig. 3d), suggesting that $U^P$47 and $G^+$15 cooperatively stabilize the tRNA core structure and contribute to cellular thermotolerance. $U^P$47 flexibly deals with the structural change due to thermal denaturation of the core structure, like a padlock, whereas $G^+$15 tightly fixes the core structure, like a screw bolt (Supplementary Note 3). On the basis of these findings, we propose a new mechanism of tRNA stabilization mediated by two distinct but concerted actions of tRNA modification.

In eukaryotic mRNAs and non-coding RNAs, $N^6$-methyladenosine ($m^6$A) has a critical role in RNA metabolism and function as a reversible RNA modification[45]. If $U^P$47 is a reversible modification, it is expected that tRNA function and stability are dynamically regulated by a writer and eraser, raising the possibility of epitranscriptomic regulation of tRNAs in translation. The mechanism closely resembles post-translational modification of proteins. Phosphorylation and dephosphorylation rapidly and dynamically control protein function[46–48]. Because tRNA is a stable molecule with a low turnover rate and long lifetime in the cell, it would be reasonable for tRNA function to be regulated by $U^P$47 modification. We found efficient dephosphorylation of $U^P$47 by TkKptA in vitro (Fig. 5b) and confirmed the in vivo activity of Tpt1/KptA homologues in *E. coli* cells (Fig. 5d, e and Extended Data Fig. 10a–d). In fact, tRNA stability is regulated by thermophile-specific tRNA modifications including $m^5s^2$U and $ac^4$C, which become much more abundant as the growth temperature increases[14,49] but are not reversible. Reversible $U^P$47 modification would be beneficial for hyperthermophilic organisms in extremely harsh environments. Future studies will be necessary to investigate $U^P$47 frequency and the expression levels of ArkI and KptA under various growth conditions, including during rapid changes in growth temperature and introduction of environmental stresses.

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

## Methods

### Archaeal strains and media

*S. tokodaii* str. 7, *Methanosarcina acetivorans* C2A and *Thermoplasma acidophilum* were kindly provided by T. Oshima (Kyowa Kako Co., Ltd), T. Yokogawa (Gifu University) and H. Hori (Ehime University), respectively. *Sulfolobus acidocaldarius* (JCM no. 8929), *Saccharolobus solfataricus* (JCM no. 8930), *Aeropyrum pernix* (JCM no. 9820), *Pyrobaculum oguniense* (JCM no. 10595) and *N. viennensis* (JCM no. 19564) were obtained from Japan Collection of Microorganisms, RIKEN BRC which is participating in the National BioResource Project of the MEXT, Japan.

*S. tokodaii* and *S. acidocaldarius* were cultured at 80 °C in JCM medium no. 165 consisting of 1 g l$^{-1}$ yeast extract, 1 g l$^{-1}$ casamino acids, 1.3 g l$^{-1}$ $(NH_4)_2SO_4$, 0.28 g l$^{-1}$ $KH_2PO_4$, 0.25 g l$^{-1}$ $MgSO_4 \cdot 7H_2O$, 0.07 g l$^{-1}$ $CaCl_2 \cdot 2H_2O$, 2.0 mg l$^{-1}$ $FeCl_3 \cdot 6H_2O$, 1.8 mg l$^{-1}$ $MnCl_2 \cdot 4H_2O$, 4.5 mg l$^{-1}$ $Na_2B_4O_7 \cdot 10H_2O$, 0.22 mg l$^{-1}$ $ZnSO_4 \cdot 7H_2O$, 0.05 mg l$^{-1}$ $CuCl_2 \cdot 2H_2O$, 0.03 mg l$^{-1}$ $Na_2MoO_4 \cdot 2H_2O$, 0.03 mg l$^{-1}$ $VOSO_4 \cdot H_2O$ and 0.01 mg l$^{-1}$ $CoSO_4 \cdot 7H_2O$ (adjusted to pH 2.5 with $H_2SO_4$). *S. solfataricus* was cultured at 80 °C in JCM medium no. 171 consisting of 1 g l$^{-1}$ yeast extract, 2.5 g l$^{-1}$ $(NH_4)_2SO_4$, 3.1 g l$^{-1}$ $KH_2PO_4$, 0.2 g l$^{-1}$ $MgSO_4 \cdot 7H_2O$, 0.25 g l$^{-1}$ $CaCl_2 \cdot 2H_2O$, 1.8 mg l$^{-1}$ $MnCl_2 \cdot 4H_2O$, 4.5 mg l$^{-1}$ $Na_2B_4O_7 \cdot 10H_2O$, 0.22 mg l$^{-1}$ $ZnSO_4 \cdot 7H_2O$, 0.05 mg l$^{-1}$ $CuCl_2 \cdot 2H_2O$, 0.03 mg l$^{-1}$ $Na_2MoO_4 \cdot 2H_2O$, 0.03 mg l$^{-1}$ $VOSO_4 \cdot H_2O$ and 0.01 mg l$^{-1}$ $CoSO_4 \cdot 7H_2O$ (adjusted to pH 4.0 with $H_2SO_4$). *A. pernix* was cultured at 90 °C in JCM medium no. 224 consisting of 1 g l$^{-1}$ yeast extract, 1 g l$^{-1}$ peptone, 1 g l$^{-1}$ $Na_2S_2O_3 \cdot 5H_2O$, 24.0 g l$^{-1}$ NaCl, 7.0 g l$^{-1}$ $MgSO_4 \cdot 7H_2O$, 5.3 g l$^{-1}$ $MgCl_2 \cdot 6H_2O$, 0.7 g l$^{-1}$ KCl and 0.1 g l$^{-1}$ $CaCl_2 \cdot 2H_2O$ (adjusted to pH 7.0 with NaOH). *P. oguniense* was cultured at 90 °C in JCM medium no. 165 with addition of 1.0 g l$^{-1}$ $Na_2S_2O_3 \cdot 5H_2O$ (adjusted to pH 7.25 with NaOH). *N. viennensis* was cultured at 42 °C in JCM medium no. 1004 consisting of 1 g l$^{-1}$ NaCl, 0.5 g l$^{-1}$ KCl, 0.4 g l$^{-1}$ $MgCl_2 \cdot 6H_2O$, 0.2 g l$^{-1}$ $KH_2PO_4$, 0.1 g l$^{-1}$ $CaCl_2 \cdot 2H_2O$, 1.0 ml l$^{-1}$ modified trace element mixture (30 mg l$^{-1}$ $H_3BO_3$, 100 mg l$^{-1}$ $MnCl_2 \cdot 4H_2O$, 190 mg l$^{-1}$ $CoCl_2 \cdot 6H_2O$, 24 mg l$^{-1}$ $NiCl_2 \cdot 6H_2O$, 2 mg l$^{-1}$ $CuCl_2 \cdot 2H_2O$, 144 mg l$^{-1}$ $ZnSO_4 \cdot 7H_2O$, 36 mg l$^{-1}$ $Na_2MoO_4 \cdot 2H_2O$ and 0.3% HCl), 1.0 ml l$^{-1}$ vitamin solution (20 mg l$^{-1}$ biotin, 20 mg l$^{-1}$ folic acid, 100 mg l$^{-1}$ pyridoxine·HCl, 50 mg l$^{-1}$ thiamine·HCl, 50 mg l$^{-1}$ riboflavin, 50 mg l$^{-1}$ nicotinic acid, 50 mg l$^{-1}$ DL-calcium pantothenate, 1 mg l$^{-1}$ vitamin $B_{12}$, 50 mg l$^{-1}$ *p*-aminobenzoic acid and 2 g l$^{-1}$ choline chloride (adjusted to pH 7.0 with KOH)), 1.0 ml l$^{-1}$ 7.5 mM EDTA·Na·Fe(III) solution (pH 7.0), 2.0 ml l$^{-1}$ 1 M $NaHCO_3$ solution, 10 ml l$^{-1}$ HEPES solution (238.4 g l$^{-1}$ HEPES (free acid) and 24 g l$^{-1}$ NaOH), 1.0 ml l$^{-1}$ 1 M $NH_4Cl$ solution and 1.0 ml l$^{-1}$ 1 M sodium pyruvate solution (adjusted to pH 7.6 with NaOH).

*T. kodakarensis* was cultured at 83 °C, 87 °C or 91 °C, in nutrient-rich medium (ASW-YT-S$^0$ or MA-YT-Pyr) or synthetic medium containing amino acids (ASW-AA-S$^0$), under strict anaerobic conditions. ASW-YT-S$_0$ medium contains 0.8× artificial sea water (ASW)[50], 10 g l$^{-1}$ yeast extract, 5.0 g l$^{-1}$ tryptone, 2.0 g l$^{-1}$ elemental sulfur and 0.1% (wt/vol) resazurin. MA-YT-Pyr medium contains 30.5 g l$^{-1}$ Marine Art SF-1 (Osaka Yakken), 10 g l$^{-1}$ yeast extract, 5.0 g l$^{-1}$ tryptone, 5.0 g l$^{-1}$ pyruvate sodium and 0.1% (wt/vol) resazurin. ASW-AA-S$^0$ medium contains 0.8× ASW, 0.5× amino acid solution[50], modified Wolfe's trace minerals (0.5 g l$^{-1}$ $MnSO_4 \cdot 2H_2O$, 0.1 g l$^{-1}$ $CoCl_2$, 0.1 g l$^{-1}$ $ZnSO_4$, 0.01 g l$^{-1}$ $CuSO_4 \cdot 5H_2O$, 0.01 g l$^{-1}$ $AlK(SO_4)_2$, 0.01 g l$^{-1}$ $H_3BO_3$ and 0.01 g l$^{-1}$ $NaMoO_4 \cdot 2H_2O$), 5.0 ml l$^{-1}$ vitamin mixture[51], 2.0 g l$^{-1}$ elemental sulfur and 0.1% (wt/vol) resazurin. For plate cultivation, 2.0 ml l$^{-1}$ polysulfide solution (20% elemental sulfur in 67% $Na_2S \cdot 9H_2O$ solution) was added instead of elemental sulfur, and the media were solidified with 1.0% Gelrite (Fujifilm Wako Pure Chemical Corporation). When *pyrF*-negative transformants were selected0, 75% 5-fluoroorotic acid (5-FOA) was added. We used ASW-YT-S$^0$ medium for standard cultivation, MA-YT-Pyr medium for growth comparisons and ASW-AA-S$^0$ medium for construction of the gene knockout strain.

### Preparation of tRNA fractions

For small-scale preparation (~100-ml culture), archaeal cells were resuspended in 3 ml solution D (4 M guanidine thiocyanate, 25 mM citrate–NaOH (pH 7.0), 0.5% (wt/vol) *N*-lauroylsarcosine sodium salt and 1 mM 2-mercaptoethanol) and mixed with an equal volume of water-saturated phenol and 1/10 volume of 3 M sodium acetate (pH 5.3). The mixture was shaken for 1 h on ice and mixed with 1/5 volume of chloroform, followed by centrifugation at 8,000*g* for 10 min at 4 °C. The supernatant was collected and mixed with an equal volume of chloroform, followed by centrifugation at 8,000*g* for 10 min at 4 °C. Total RNA was obtained from the resultant supernatant by isopropanol precipitation. The total RNA prepared in this manner was separated by 10% denaturing PAGE, followed by staining with SYBR Gold or toluidine blue. The visualized tRNA fraction including class I and class II tRNAs was cut out and eluted from the gel slice with elution buffer (0.3 M sodium acetate (pH 5.3) and 0.1% (wt/vol) SDS), followed by filtration to remove the gel pieces and ethanol precipitation for RNA-MS analysis of the tRNA fraction.

For large-scale preparation of tRNA fractions from *S. tokodaii*, cell pellets (53 g) were resuspended in 530 ml solution D and then mixed with 53 ml of 3 M sodium acetate (pH 5.3) and 425 ml neutralized phenol. The mixture was shaken for 1 h on ice to which 106 ml chloroform/isoamyl alcohol (49:1) was added, followed by centrifugation at 4,500*g* for 20 min at 4 °C. The supernatant was collected and mixed with 106 ml chloroform/isoamyl alcohol (49:1), followed by centrifugation at 4,500*g* for 15 min at 4 °C. The aqueous phase was collected and then subjected to isopropanol precipitation. The collected RNA was resuspended in 53 ml water and mixed with 80 ml TriPure Isolation Reagent (Roche), followed by centrifugation at 10,000*g* for 20 min at 4 °C. The supernatant was collected and mixed with 36 ml chloroform/isoamyl alcohol (49:1), followed by centrifugation at 10,000*g* for 10 min at 4 °C. The aqueous phase was collected and precipitated with isopropanol. The prepared total RNA (608 mg) was dissolved in 250 ml of buffer consisting of 20 mM HEPES-KOH (pH 7.6), 200 mM NaCl and 1 mM DTT and then loaded on a DEAE Sepharose Fast Flow column (320-ml beads) and fractionated with a gradient of NaCl from 200 to 500 mM. Fractions containing tRNA were collected by isopropanol precipitation.

### Isolation of individual tRNAs

Isolation of individual tRNAs from thermophilic organisms is extremely difficult owing to their high melting temperatures, which are the consequence of their high G+C content and complex modifications. We thus optimized our original method for RNA isolation by RCC[24] or chaplet column chromatography (CCC)[52]. Approximately 200 absorbance at 260 nm ($A_{260}$) units of the *S. tokodaii* tRNA fraction was subjected to RCC. The isolation procedure was carried out as follows: hybridization at 66 °C in 6× NHE buffer (30 mM HEPES-KOH (pH 7.5), 15 mM EDTA (pH 8.0), 1.2 M NaCl, 1 mM DTT), washing at 50 °C with 0.1× NHE buffer (0.5 mM HEPES-KOH (pH 7.5), 0.25 mM EDTA (pH 8.0), 20 mM NaCl, 0.5 mM DTT) and elution at 72 °C with 0.1× NHE buffer. Eluted tRNAs were recovered by ethanol precipitation. Mature and precursor tRNAs were separated by 10% denaturing PAGE and stained with SYBR Gold. Visualized bands of mature and precursor tRNAs were cut out and eluted from the gel slices with elution buffer, followed by filtration to remove the gel pieces and precipitation with ethanol.

To crystalize native tRNA bearing U$^P$47, we conducted large-scale isolation of *S. tokodaii* tRNA$^{Val3}$ using CCC[52]. The *S. tokodaii* tRNA fraction (2,000 $A_{260}$ units) was subjected to CCC with tandem affinity chaplet columns for tRNA$^{Val3}$, tRNA$^{Ile2}$ and tRNA$^{Phe}$. The isolation procedure was carried out as follows: hybridization at 66 °C in 6× NHE buffer, washing separately at 50 °C with 0.1× NHE buffer and elution at 72 °C with 0.1× NHE buffer. The eluted tRNAs were recovered by isopropanol precipitation. The sequences of the DNA probes are shown in Supplementary Table 6. The isolated tRNA$^{Val3}$ was further purified by anion exchange chromatography to completely remove tRNA$^{Val2}$, as described below.

## RNA mass spectrometry

For tRNA fragment analysis by RNA-MS, 30 ng (900 fmol) of the isolated tRNA or 150 ng (4.5 pmol) of tRNA mixture was digested with RNase $T_1$ (Epicentre or Thermo Fisher Scientific) or RNase A (Ambion) and analysed with a linear ion trap–Orbitrap hybrid mass spectrometer (LTQ Orbitrap XL, Thermo Fisher Scientific) equipped with a custom-made nanospray ion source and a splitless nanoHPLC system (DiNa, KYA Technologies) as described previously[26,27]. To analyse $\Psi$ sites, tRNA was treated with acrylonitrile to cyanoethylate $\Psi$[53] and subjected to RNA-MS. For dephosphorylation of the $U^p47$-containing fragment (Extended Data Fig. 4a, b), RNase $T_1$ digestion was performed in the presence of 0.01 U $\mu l^{-1}$ bacterial alkaline phosphatase (BAP C75, Takara Bio). To precisely map tRNA modifications, RNA fragments were decomposed by CID in the instrument. The normalized collision energy of LTQ Orbitrap XL was set to 40%. Mongo Oligo Mass Calculator v2.08 (https://mods.rna.albany.edu/masspec/Mongo-Oligo) was used for assignment of the product ions in CID spectra.

For nucleoside analysis, 800 ng (24 pmol) of the isolated tRNA$^{Val3}$ was digested with 0.09 U nuclease $P_1$ (Fujifilm Wako Pure Chemical Corporation) in 20 mM ammonium acetate (pH 5.2) at 50 °C for 1 h and mixed with 1/8 volume of 1 M trimethylamine-HCl (TMA-HCl) (pH 7.2) and 0.06 U phosphodiesterase I (Worthington Biochemical Corporation), followed by incubation at 37 °C for 1 h. To this mixture, 0.08 U BAP was added, and the sample was incubated at 50 °C for 1 h. After that, 9 volumes of acetonitrile were added, followed by LC–MS/MS analysis as described in refs. [25,54] with some modifications as follows. The samples were chromatographed with a ZIC-cHILIC column (3-μm particle size, 2.1 × 150 mm; Merck) and eluted with 5 mM ammonium acetate (pH 5.3) (solvent A) and acetonitrile (solvent B) at a flow rate of 100 μl $min^{-1}$ with a multistep linear gradient: 90–50% solvent B for 30 min, 50% solvent B for 10 min, 50–90% solvent B for 5 min and then initialization with 90% solvent B. The chromatographed eluent was directly introduced into the electrospray ionization source of the Q Exactive Hybrid Quadrupole–Orbitrap mass spectrometer (Thermo Fisher Scientific).

For nucleotide analysis, 800 ng (24 pmol) of the tRNA fraction or individual tRNA was digested with 0.09 U nuclease $P_1$ in 20 mM ammonium acetate (pH 5.2) at 50 °C for 1 h and then mixed with 9 volumes of acetonitrile for LC–MS. The digests were chromatographed with a ZIC-cHILIC column and analysed by Q Exactive Hybrid Quadrupole–Orbitrap mass spectrometer (Thermo Fisher Scientific) or LTQ Orbitrap XL (Thermo Fisher Scientific) with a multistep linear gradient: 90–50% solvent B for 30 min, 50% solvent B for 10 min, 50–90% solvent B for 5 min and then initialization with 90% solvent B.

The acquired LC–MS data were analysed using Xcalibur 4.1 (Thermo Fisher Scientific) and were visualized with Canvas X (Nihon poladigital k.k.).

## Isolation and detection of pN$^{324}$p

Five $A_{260}$ units of the *S. tokodaii* tRNA fraction was completely digested with nuclease $P_1$. Digests containing pN$^{324}$m$^5$C dinucleotide were subjected to periodate oxidation with 10 mM NaIO$_4$ for 1 h on ice in the dark. The reaction was stopped by addition of 1 M L-rhamnose and incubation for 30 min. For β-elimination, an equal volume of 2 M lysine-HCl (pH 8.5) was added, and the sample was incubated at 45 °C for 90 min. The product containing pN$^{324}$p was then subjected to anion exchange chromatography with a Q Sepharose Fast Flow column (GE Healthcare) equilibrated with 20 mM triethylammonium bicarbonate (TEAB) (pH 8.2). The eluate with 2 M TEAB was collected and dried by evaporation in vacuo. The pellet was dissolved with water and mixed with an equal volume of chloroform, followed by centrifugation at 20,000$g$ for 5 min at 4 °C. The supernatant was recovered and dried again. This process was repeated five times. The resultant digest was mixed with 9 volumes of acetonitrile and subjected to LC–MS/MS using

an LCQ-Advantage ion trap mass spectrometer (Thermo Scientific), equipped with an electrospray ionization source and an HP1100 LC system (Agilent Technologies). For LC, the digest was chromatographed with a ZIC-HILIC column (3.5 μm; pore size, 100 Å; internal diameter, 2.1 × 150 mm; Merck) and eluted with 5 mM formic acid (pH 3.4) (solvent A) and acetonitrile (solvent B) at a flow rate of 100 μl $min^{-1}$ with a multistep gradient: 90–70% solvent B for 25 min, 70–10% solvent B for 15 min, 10% solvent B for 5 min and then initialized with 90% solvent B.

## Expression and purification of recombinant proteins

Synthetic genes for *arkI* from *T. kodakarensis*, *Methanocaldococcus fervens*, *P. oguniense*, *Aquifex aeolicus*, *Nautilia profundicola* and *Leptolyngbya* sp. PCC7376 were designed with codons optimized for *E. coli* expression and synthesized by GENEWIZ or Thermo Fisher Scientific. Each gene was cloned into the pE-SUMO-TEV vector by the SLiCE method[55]. *N. viennensis arkI* was PCR amplified from genomic DNA with a set of primers (Supplementary Table 6) and cloned into the BamHI and NotI sites of pE-SUMO-TEV.

*E. coli* BL21(DE3) or Rosetta2(DE3) cells transformed with the pE-SUMO-TEV vector carrying each *arkI* gene were cultured in 250 ml or 1 l of LB containing 50 μg $ml^{-1}$ kanamycin and 20 μg $ml^{-1}$ chloramphenicol when necessary. His$_6$–SUMO-tagged recombinant protein was expressed at 37 °C for 3–4 h by induction with 0.1 or 1 mM IPTG or 2% (wt/vol) lactose when the cells reached OD$_{610}$ = 0.4–0.6. *P. oguniense* ArkI was expressed in cells cultured overnight at 18 °C. The collected cells were resuspended in lysis buffer (50 mM HEPES-KOH (pH 8.0), 150 mM KCl, 2 mM MgCl$_2$, 20 mM imidazole, 12% (vol/vol) glycerol, 1 mM 2-mercaptoethanol and 1 mM PMSF) and disrupted by sonication, followed by centrifugation at 15,000$g$ for 15 min at 4 °C. The supernatant was boiled at 60 °C for 20 min (for ArkI homologues from *T. kodakarensis*, *M. fervens*, *P. oguniense* and *A. aeolicus*) and centrifuged at 15,000$g$ for 15 min at 4 °C. The recombinant protein was affinity captured on an Ni-Sepharose 6 Fast Flow column (GE Healthcare) and then eluted with lysis buffer containing 300 mM imidazole, followed by gel filtration with a PD-10 column (GE Healthcare) to remove the imidazole. The recombinant protein for *N. viennensis* ArkI was purified using a HisTrap column (GE Healthcare) with a linear gradient of 0–500 mM imidazole, followed by dialysis using a Slide-A-Lyzer Dialysis Cassette (Thermo Fisher Scientific) to remove imidazole. The purified protein was subjected to Ulp1 digestion at 4 °C overnight to cleave the His$_6$–SUMO tag and then passed through a Ni-Sepharose 6 Fast Flow column to remove the tag. Because ArkI homologues from *M. fervens* (MfArkI) and *Leptolyngbya* sp. PCC7376 (LeArkI) aggregated following tag removal, His$_6$–SUMO tag-fused proteins of these homologues were used for the phosphorylation assay. Purified protein was quantified by the Bradford method using BSA as a standard.

For large-scale preparation of *T. kodakarensis* ArkI for crystallization, the *E. coli* BL21(DE3) strain carrying pE-SUMO-TkArkI was cultured in 2 l of LB containing 50 μg $ml^{-1}$ kanamycin and TkArkI was expressed at 25 °C overnight by induction with 0.1 mM IPTG when the cells reached OD$_{610}$ = 0.4. The cells were collected and disrupted by sonication in lysis buffer (50 mM HEPES-KOH (pH 8.0), 150 mM KCl, 2 mM MgCl$_2$, 20 mM imidazole, 12% (vol/vol) glycerol, 1 mM 2-mercaptoethanol and 1 mM PMSF). The protein was purified using a HisTrap column with a linear gradient of 20–520 mM imidazole. Fractions containing TkArkI were pooled and subjected to Ulp1 digestion at 4 °C overnight to cleave the tag, followed by passage through a Ni-Sepharose 6 Fast Flow column to remove the tag fragment. The flow-through fraction was filtered through a 0.45-μm PVDF membrane to remove the resin. The protein was further purified by affinity chromatography with a HiTrap Heparin HP column (GE Healthcare) using a linear gradient of 150–1,150 mM KCl. TkArkI was further purified by size exclusion chromatography using a Superdex 75 10/300 GL column (GE Healthcare) with buffer containing 20 mM Tris-HCl (pH 8.0), 150 mM NaCl and 10 mM 2-mercaptoethanol and then concentrated to 5.74 mg $ml^{-1}$ and stored at −80 °C.

The *T. kodakarensis kptA* gene was PCR amplified from genomic DNA from *T. kodakarensis* with the primers listed in Supplementary Table 6 and cloned into pE-SUMO-TEV to give pE-SUMO-TEV-*tkkptA*. The *E. coli* Rosetta2(DE3) strain carrying pE-SUMO-TEV-*tkkptA* was cultured in 1 l LB containing 50 μg ml$^{-1}$ kanamycin and 20 μg ml$^{-1}$ chloramphenicol, and TkKptA was expressed at 37 °C for 3 h by induction with 0.1 mM IPTG when the cells reached OD$_{610}$ = 0.6. The recombinant TkKptA was purified as described above. The gene encoding Tpt1p was PCR amplified from the genomic DNA of *S. cerevisiae* BY4742 with the set of primers listed in Supplementary Table 6 and was cloned into pET21b (Merck) between the NdeI and XhoI sites. Recombinant Tpt1p was purified as described above.

## Removal of the 2′-phosphate of U$^p$47 by Tpt1p

Removal of the 2′-phosphate of U$^p$47 by yeast Tpt1p was performed as described[33]. Individual tRNAs or the tRNA fraction was incubated for 3 h at 30 °C in a reaction mixture (25 μl) consisting of 20 mM Tris-HCl (pH 7.4), 0.5 mM EDTA (pH 8.0), 1 mM NAD$^+$, 2.5 mM spermidine, 0.1 mM DTT, 0.9 μM tRNA and 0.1 μg μl$^{-1}$ recombinant Tpt1p. The tRNA was extracted by phenol/chloroform treatment and recovered by ethanol precipitation, followed by desalting with Centri-Sep spin columns (Princeton Separations). For crystallization of Tpt1p-treated tRNA, *S. tokodaii* tRNA$^{Val3}$ (202.5 μg) was dephosphorylated by yeast Tpt1p in a 200-μl reaction mixture.

## Measurement of the thermal stability of tRNA

*S. tokodaii* tRNA$^{Val3}$ (25 pmol) with or without U$^p$47 was dissolved in degassed buffer consisting of 50 mM Tris-HCl (pH 7.4), 100 mM NaCl and 1 mM MgCl$_2$ and incubated at 80 °C for 5 min, followed by cooling to 25 °C at a rate of 0.1 °C s$^{-1}$. The samples were placed onto a Type 8 multi-micro UV quartz cell (path length, 10 mm). The hyperchromicity of tRNA was monitored on a UV–visible light spectrophotometer (V-630, JASCO). The gradients were as follows: 25 °C for 30 s, 25–40 °C at 5 °C min$^{-1}$, 40 °C for 5 min and 40–105 °C at 0.5 °C min$^{-1}$. The $T_m$ was calculated using Spectra Manager v2 (JASCO). Melting curves were generated using Microsoft Excel.

## RNase probing of tRNA

*S. tokodaii* tRNA$^{Val3}$ (25 pmol) with or without U$^p$47 was labelled with $^{32}$P at the 3′ terminus by ligation with [5′-$^{32}$P]cytidine 3′,5′-bisphosphate (PerkinElmer). The labelled tRNA was separated on a 7.5% (wt/vol) polyacrylamide gel containing 7 M urea, 1× TBE and 10% (vol/vol) glycerol and was purified by gel extraction. Labelled tRNA was mixed with the *S. tokodaii* tRNA fraction as a carrier to a concentration of 100,000 counts per minute (c.p.m.) per $A_{260}$ unit and was precipitated with ethanol. The pellet was dissolved in water to a concentration of 0.1 $A_{260}$ units per μl. For the RNase degradation assay, the labelled tRNA (0.1 $A_{260}$ units, 10,000 c.p.m.) was incubated at 65 °C in a reaction mixture consisting of 10 mM HEPES-KOH (pH 7.6), 0.5 mM MgCl$_2$, 100 mM NaCl and 0.1 U μl$^{-1}$ RNase I (Promega). At time points of 1, 3, 5, 10, 15 and 30 min after starting the reaction, aliquots were taken from the mixture and mixed well with chilled phenol/chloroform/isoamyl alcohol (25:24:1, pH 7.9) to stop the reaction, followed by centrifugation at 15,000 g for 15 min at 4 °C. The supernatant was collected and treated with an equal volume of chloroform, followed by centrifugation at 15,000 g for 5 min at 4 °C. The supernatant was mixed with 2× loading solution (2× TBE, 7 M urea, 13.33% (wt/vol) sucrose, 0.05% (wt/vol) xylene cyanol and 0.05% (wt/vol) bromophenol blue) and subjected to 10% denaturing PAGE. The gel was exposed to an imaging plate, and radioactivity was visualized by using an FLA-7000 imaging analyser (Fujifilm). Graphs were generated using Microsoft Excel.

## Crystallization of *S. tokodaii* tRNA$^{Val3}$

*S. tokodaii* tRNA$^{Val3}$ (500 μg), isolated as described above, was refolded in annealing buffer (50 mM HEPES-KOH (pH 7.6), 5 mM MgCl$_2$ and 1 mM DTT) by incubation for 5 min at 80 °C and cooling to 25 °C with a rate of 0.1 °C s$^{-1}$. tRNA$^{Val3}$ was further purified by anion exchange chromatography using a Mono Q 5/50 GL column (GE Healthcare) with a linear gradient of 200–1,000 mM NaCl. The major peak was collected, precipitated with isopropanol, dissolved in water and precipitated with ethanol. Tpt1p-treated tRNA$^{Val3}$ was prepared with the same procedure as described above. The purified tRNA was dissolved in buffer consisting of 10 mM Tris-HCl (pH 7.1) and 5 mM MgCl$_2$ to a concentration of 50 μM. One microlitre of tRNA solution was mixed with 1 μl Natrix 2 no. 32 (80 mM NaCl, 12 mM spermine-4HCl, 40 mM sodium cacodylate·3H$_2$O (pH 7.0) and 30% (vol/vol) MPD) (Hampton Research) on silicon-coated glass and crystalized by the hanging drop vapor diffusion method at 20 °C.

## Crystallization of *T. kodakarensis* ArkI

The concentration of TkArkI was adjusted to 5 mg ml$^{-1}$ before crystallization. One microlitre of the protein solution was mixed with 0.5 μl reservoir solution, containing 25% (vol/vol) ethylene glycol. TkArkI was crystallized by the hanging drop vapor diffusion method at 20 °C.

## Data collection and crystal structure determination

The datasets were collected at beamline BL-17A at the Photon Factory at KEK, Japan. For data collection for the tRNA$^{Val3}$ crystals, the crystals were cryoprotected with a portion of the reservoir solution. For data collection for the native TkArkI crystal, the crystal was cryoprotected with solution containing 25% (vol/vol) ethylene glycol, 2 mM MgCl$_2$ and 1 mM ATP. For data collection for the iodide-derivative TkArkI crystal, the crystal was briefly soaked in and cryoprotected with solution containing 300 mM potassium iodide and 22.5% (vol/vol) ethylene glycol, and the diffraction dataset was collected at a wavelength of 1.5 Å. The datasets were indexed, integrated and scaled using xds[56]. The initial phase of tRNA$^{Val3}$ was determined by molecular replacement with Phaser[57]. The structure of *T. thermophilus* tRNA$^{Val}$ (PDB, 1IVS)[58] was used for the model. The initial phase of TkArkI was determined by the SAD method using the anomalous signal of iodide ions. The iodine sites were located by SHELX[59], and the initial phase was calculated by Phaser. Subsequent density modification and initial model building were performed with RESOLVE[60]. The model was further modified with Coot[61] and refined with Phenix[62]. Crystal structures and their electron density maps were visualized using PyMOL, Cuemol or Coot. Torsion angles of the tRNAs were analysed with DSSR software[63].

## Analysis of ligands bound to TkArkI

TkArkI purified by affinity chromatography with a HiTrap Heparin HP column (GE Healthcare) (100 pmol) was mixed with [$^{15}$N]adenosine (10 pmol) and [$^{15}$N]guanosine (10 pmol) as tracer molecules, followed by addition of 4 volumes of methanol, an equal volume of chloroform and 3 volumes of water and vigorous mixing. The denatured protein was removed by centrifugation at 15,000 g for 1 min at 4 °C. The supernatant was dried in vacuo and dissolved in 20 μl water. Half of the extract was analysed by LC–MS. The tracer molecules were prepared by dephosphorylation of [$^{15}$N]ATP and [$^{15}$N]GTP as follows: 1,000 pmol each of [$^{15}$N]ATP (Silantes) and [$^{15}$N]GTP (Silantes) was treated with 0.04 U alkaline phosphatase (PAP, from *Shewanella* sp. SIB1, BioDynamics Laboratory) in 20 mM ammonium acetate (pH 8.0) at 60 °C for 30 min. After dephosphorylation, PAP was heat denatured at 95 °C for 5 min.

## Construction of gene knockout strains of *T. kodakarensis*

Knockout strains of *T. kodakarensis* were constructed by pop-in/pop-out recombination as described previously[64]. The 5′ and 3′ flanking regions (about 1,000 bp) of *T. kodakarensis arkI* and *kptA* were PCR amplified from genomic DNA with a set of primers (Supplementary Table 6) and inserted into the pUD3 vector bearing the *pyrF* marker[65] to yield pUD3-*arkI* and pUD3-*kptA*. The *T. kodakarensis* KU216 strain (Δ*pyrF*) was transformed with pUD3-*arkI* or pUD3-*kptA*, and the

uracil-prototrophic transformants generated by pop-in recombination were selected on an ASW-AA-S$^0$ plate without uracil. The selected strains were then cultured on an ASW-AA-S$^0$ plate supplemented with 5-FOA to obtain uracil-auxotrophic, 5-FOA-resistant transformants formed by pop-out recombination. The knockout strains of *arkI* or *kptA* were selected among the transformants by genomic PCR with a set of primers (Supplementary Table 6). The double-knockout strain of *arkI* and *queE* (Δ*arkI*/*queE*::Tn) was constructed by deletion of *arkI* from FFH05 (*queE*::Tn) isolated from a random mutagenesis library[19]. *T. kodakarensis* strains used in this study are listed in Supplementary Table 7.

### Growth phenotype analysis

*T. kodakarensis* KU216 (wild type), FFH05 (*queE*::Tn), Δ*arkI* and Δ*arkI*/*queE*::Tn strains were precultured in MA-YT-Pyr medium at 83 °C overnight and inoculated into 8 ml fresh MA-YT-Pyr medium with an initial $OD_{600}$ of 0.01. The cells were cultured at 83 °C, 87 °C or 91 °C, and cell growth was monitored every 2 h by measuring $OD_{600}$ with an S1200 diode array spectrophotometer. Graphs were generated using Microsoft Excel.

### In vitro transcription of tRNA

For in vitro transcription of *T. kodakarensis* tRNA$^{Val3}$ and its G5–C68 variants by T7 RNA polymerase[66], template DNAs were constructed by PCR using synthetic DNA (Supplementary Table 6). The tRNAs were transcribed at 37 °C overnight in a reaction mixture consisting of 40 mM Tris-HCl (pH 7.5), 24 mM $MgCl_2$, 5 mM DTT, 2.5 mM spermidine, 0.01% (vol/vol) Triton X-100, 0.8 μg ml$^{-1}$ T7 RNA polymerase, 1 μg ml$^{-1}$ pyrophosphatase, 30 nM DNA template, 2 mM ATP, 2 mM CTP, 2 mM UTP, 2 mM GTP and 10 mM GMP, followed by extraction with phenol/chloroform treatment and desalting with PD-10 columns (GE Healthcare). In vitro transcripts prepared in this way were separated by 10% denaturing PAGE, followed by staining with toluidine blue. The stained bands were cut out and eluted from the gel slice with elution buffer, followed by filtration to remove the gel pieces and ethanol precipitation.

### In vitro phosphorylation of tRNA by ArkI

U$^p$47 formation by TkArkI was carried out at 70 °C for 20 min in a reaction mixture (30 μl) containing 50 mM HEPES-KOH (pH 7.5), 1 mM $MgCl_2$, 1 mM $MnCl_2$, 1 mM DTT, 10% (vol/vol) glycerol, 0.5 mM ATP, 0.9 μM tRNA fraction (from the *T. kodakarensis* Δ*arkI* strain) and 1 μM TkArkI. After the reaction, the tRNA was extracted by acidic phenol/chloroform, desalted on a NAP-5 column (GE Healthcare) and precipitated with isopropanol. For RNA-MS, the prepared tRNA was dialysed against water on a nitrocellulose membrane (0.025-μm VSWP, MF-Millipore, Merck) for 2 h (drop dialysis). To examine GTP as a phosphate donor, 0.5 mM ATP or GTP was added to the reaction mixture and U$^p$47 formation was performed with 0.5 μM TkArkI for 5 min, followed by RNA-MS analysis. The activities of TkArkI variants were measured by γ-phosphate transfer from [γ-$^{32}$P]ATP to tRNA similarly to the kinetic studies of TkArkI (see below). tRNA phosphorylation was performed at 70 °C for 15 min in an 8-μl reaction mixture. For PAGE analysis, 4 μl of the reaction mixture was mixed with 4 μl of 2× loading solution, resolved by 10% denaturing PAGE and exposed to an imaging plate to visualize radiolabelled RNA with an FLA-9000 imaging analyser (Fujifilm). The gel image was analysed using Multi Gauge (Fujifilm). Bar graphs with independent plots were prepared with R (R Foundation). For phosphorylation of total RNA, the reaction was performed at 70 °C for 30 min in an 8-μl reaction mixture consisting of 50 mM HEPES-NaOH (pH 7.5), 1 mM $MgCl_2$, 1 mM $MnCl_2$, 1 mM DTT, 10% (vol/vol) glycerol, 100 μM [γ-32P] ATP (3,000 mCi mmol−1; PerkinElmer), 1.8 μM TkArkI and 50 ng μl$^{-1}$ total RNA fraction (from the *T. kodakarensis* Δ*arkI* strain). Then, 0.5 μl of 50 mM EDTA (pH 8.0) was added, and 4 μl of reaction mixture was mixed with 2× loading solution, resolved by 10% denaturing PAGE and visualized as described above.

Formation of U$^p$47 by other ArkI homologues was carried out at 70 °C for 30 min in a reaction mixture (30 μl) containing 50 mM PIPES-NaOH (pH 6.9), 125 mM NaCl, 1 mM $MgCl_2$, 1 mM $MnCl_2$, 1 mM DTT, 10% (vol/vol) glycerol, 500 μM ATP, 0.05 mg ml$^{-1}$ BSA (Takara), 1 μM tRNA transcript and 0.5 μM ArkI protein. For NvArkI, the reaction temperature was set to 45 °C. For ArkI homologue from *N. profundicola* (NpArkI), the reaction was carried out at 50 °C for 60 min. After the reaction, tRNA was prepared as described above. For PAGE analysis, U$^p$47 formation was carried out in a reaction mixture (8 μl) containing 50 mM PIPES-NaOH (pH 6.9), 125 mM NaCl, 1 mM $MgCl_2$, 1 mM $MnCl_2$, 1 mM DTT, 10% (vol/vol) glycerol, 100 μM [γ-32P]ATP (3,000 mCi mmol−1; PerkinElmer), 0.1 mg ml$^{-1}$ BSA (Takara), 0.75 μM recombinant ArkI homologue (NpArkI, NvArkI or LeArkI) and 50 ng μl−1 *E. coli* total RNA. Then, the reaction mixture was mixed with 2× loading solution, resolved by 10% denaturing PAGE and visualized as described above.

### In vitro dephosphorylation of tRNA by *T. kodakarensis* KptA

Dephosphorylation of U$^p$47 by TkKptA was carried out at 60 °C for 1 h in a reaction mixture (30 μl) containing 20 mM Tris-HCl (pH 7.4), 0.5 mM EDTA (pH 8.0), 1 mM NAD$^+$, 2.5 mM spermidine, 0.1 mM DTT, 0.9 μM *T. kodakarensis* tRNA fraction and 0.1 μg μl$^{-1}$ recombinant TkKptA. After the reaction, the tRNA was extracted by acidic phenol/chloroform, desalted on a NAP-5 column (GE Healthcare) and precipitated with isopropanol. For RNA-MS, the prepared tRNA was desalted by drop dialysis as described above.

### Kinetic studies of *T. kodakarensis* ArkI and KptA

TkArkI-mediated U$^p$47 formation was quantified by γ-phosphate transfer from [γ-$^{32}$P]ATP to tRNA. For kinetic measurement of the tRNA substrate, tRNA phosphorylation was performed at 70 °C in a reaction mixture (25 μl) consisting of 50 mM PIPES-NaOH (pH 6.9), 125 mM NaCl, 1 mM $MgCl_2$, 1 mM $MnCl_2$, 1 mM DTT, 10% (vol/vol) glycerol, 100 μM [γ-$^{32}$P]ATP (1,500 mCi mmol$^{-1}$; PerkinElmer), 0.05 mg ml$^{-1}$ BSA (Takara), 0.05 μM TkArkI and 0.1–5.0 μM of in vitro-transcribed *T. kodakarensis* tRNA$^{Val3}$. For kinetic measurement of the ATP substrate, the ATP concentration was altered from 15.6 to 1,000 μM [γ-32P]ATP (750 mCi mmol−1; PerkinElmer) and the tRNA concentration was increased to 1.0 μM. At each time point (2 and 5 min), 8-μl aliquots were taken and mixed with an equal volume of 2× loading solution (7 M urea, 0.2% (wt/vol) bromophenol blue, 0.2% (wt/vol) xylene cyanol and 50 mM EDTA (pH 8.0)) to quench the reaction. Each sample was subjected to 10% denaturing PAGE. The gel was exposed on an imaging plate to measure radiolabelled tRNAs using an FLA-9000 imaging analyser. Kinetic parameters were calculated using Prism 7 (GraphPad).

TkKptA-mediated dephosphorylation of U$^p$47 was quantified by measuring the reduction in radioactivity for tRNA. In vitro-transcribed *T. kodakarensis* tRNA$^{Val3}$ was phosphorylated by TkArkI with [γ-$^{32}$P]ATP as described above and then purified by gel extraction and isopropanol precipitation. In addition, the same tRNA was phosphorylated by TkArkI with unlabelled ATP. By mixing labelled and unlabelled tRNAs, the specific activity of the labelled tRNA was adjusted to 6,250 c.p.m. per pmol in buffer consisting of 50 mM HEPES-KOH (pH 7.6), 5 mM $MgCl_2$ and 1 mM DTT. The labelled tRNA was incubated at 80 °C for 5 min and then cooled at room temperature, followed by isopropanol precipitation. The labelled tRNA was dissolved in water to a concentration of 8 μM (50,000 c.p.m. per μl). Dephosphorylation of the labelled tRNA by TkKptA was performed at 70 °C in a reaction mixture (30 μl) consisting of 50 mM PIPES-NaOH (pH 6.9), 125 mM NaCl, 1 mM $MgCl_2$, 1 mM $MnCl_2$, 1 mM DTT, 10% (vol/vol) glycerol, 1 mM NAD$^+$, 0.05 mg ml$^{-1}$ BSA (Takara), 1 nM TkKptA and 12.5–800 nM $^{32}$P-labelled tRNA. At each time point (2 and 5 min), 8-μl aliquots were spotted on Whatman 3MM filter paper, which was immediately soaked in 5% (wt/vol) trichloroacetic acid. The filter paper was washed three times for 15 min with ice-cold 5% (wt/vol) trichloroacetic acid, rinsed for 5 min with ice-cold ethanol and dried in air. Radioactivity on the filter paper was measured by liquid scintillation

counting (Tri-Carb 2910TR, PerkinElmer). Kinetic parameters were calculated using Prism 7.

## In vivo dephosphorylation of U$^p$47 by KptA

*N. viennensis arkI* was PCR amplified and cloned into pMW118 (Invitrogen) under the control of the synthetic constitutive J23106 promoter[67,68], followed by insertion of sequences encoding a His$_6$ tag and a 3×Flag tag at the C terminus of the *N. viennensis arkI* gene, yielding pMW-J23106-*nvarkI* (Supplementary Table 7). *T. kodakarensis kptA*, *E. coli kptA* and *S. cerevisiae tpt1* were PCR amplified and cloned into pQE-80L (Qiagen). The ampicillin resistance cassette (Amp$^r$) was replaced with a chloramphenicol resistance cassette (Cam$^r$), yielding pQE-80LC-*tkkptA*, pQE-80LC-*eckptA* and pQE-80LC-*sctpt1*, respectively (Supplementary Table 7). The *E. coli ΔtrmBΔtapT* (Kan$^r$) strain was transformed with pMW-J23106-*nvarkI* and further transformed with pQE-80LC-*tkkptA*, pQE-80LC-*eckptA* or pQE-80LC-*sctpt1*. The transformants were inoculated in 3 ml LB supplemented with 20 μg ml$^{-1}$ chloramphenicol, 50 μg ml$^{-1}$ kanamycin and 100 μg ml$^{-1}$ ampicillin and cultured at 37 °C until mid-log phase. When the OD$_{610}$ reached 0.6, IPTG was added to a final concentration of 10 or 100 μM to induce expression of the KptA/Tpt1p homologue and cells were cultured for 3.5 h. A 1.5-ml aliquot of the culture was taken, and the tRNA fraction was extracted and analysed by shotgun RNA-MS as described above. Primers, *E. coli* strains and plasmids used are listed in Supplementary Tables 6, 7. Bar graphs with independent plots were prepared with R (R Foundation).

## Drawing of chemical structures

Chemical structures were drawn with chemical structure drawing tools, including ACD/ChemSketch (ACD/Labs) or ChemDraw (PerkinElmer).

## Reporting summary

Further information on research design is available in the Nature Research Reporting Summary linked to this paper.

## Data availability

Coordinates and structure factors have been deposited in the Protein Data Bank under accession codes 7VNV, 7VNW and 7VNX. Source data are provided with this paper.

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

**Acknowledgements** We thank the members of the Suzuki laboratory for their continuous technical assistance and fruitful discussion. We also thank the beamline staff at BL-17A of the Photon Factory for technical assistance during data collection. *S. tokodaii*, *M. acetivorans* C2A, *T. acidophilum* and the *E. coli ΔtrmBΔtapT* strain were kindly provided by T. Ohshima (Kyowa Kako Co., Ltd), T. Yokogawa (Gifu University), H. Hori (Ehime University) and M. Takakura (Suzuki laboratory), respectively. This work was carried out with the support of the Isotope Science Center, University of Tokyo. This work was supported by Grants-in-Aid for Scientific Research on Priority Areas from the Ministry of Education, Science, Sports and Culture of Japan; Research Fellowships for Young Scientists from the Japan Society for the Promotion of Science (26113003, 26220205 and 18H05272 to T.S.; 26113002 and 18H03980 to K.T.; 26840005 and 17H04997 to T.O.; and 19J20723 to K. Minowa); and Exploratory Research for Advanced Technology (ERATO, JPMJER2002 to T.S.) from the Japan Science and Technology Agency (JST).

**Author contributions** K. Minowa, T.O. and K.S. mainly performed the series of experiments. K.S. and T.O. conducted LC–MS analyses and biochemical and thermodynamic analyses of *S. tokodaii* tRNA assisted by K. Miyauchi and Y.S. Crystal structure analysis of tRNA was performed by K.S., T.O., S.Y. and K. Minowa with support from K.T. Gene identification by comparative genomics was conducted by K. Minowa and T.O. Biochemical characterization of ArkI proteins was performed by K. Minowa and R.N. K. Minowa performed genetic work assisted by A.K., I.O. and T.F. K. Minowa performed structural studies of ArkI assisted by S.Y. and K.T. K. Minowa conducted *E. coli* experiments assisted by K. Miyauchi and Y.S. All authors discussed the results and revised the manuscript. T.O., K. Minowa and T.S. designed the studies and wrote the manuscript. T.S. supervised the project.

**Competing interests** The authors declare no competing interests.

**Additional information**
**Correspondence and requests for materials** should be addressed to Takayuki Ohira, Kozo Tomita or Tsutomu Suzuki.

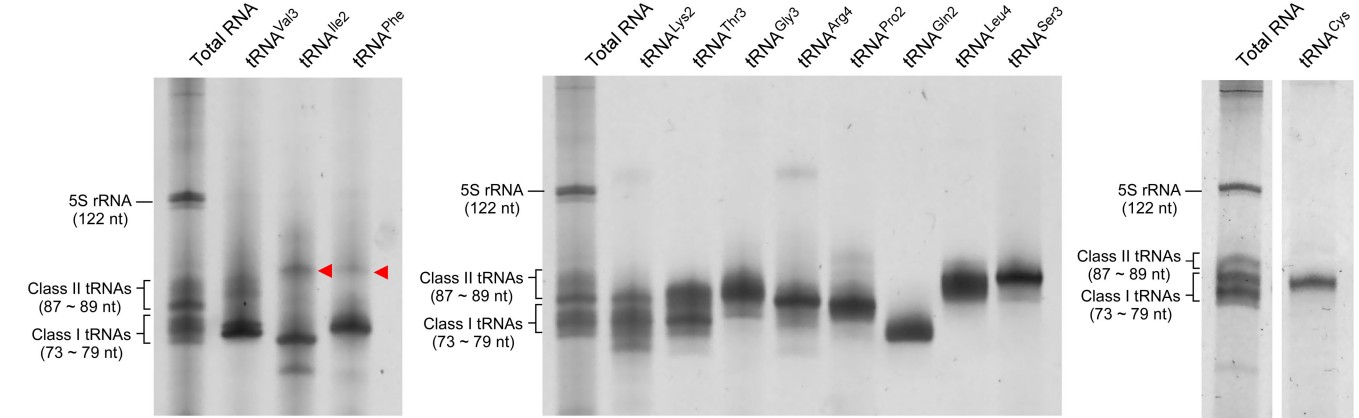

**Extended Data Fig. 1 | Isolation of *S. tokodaii* tRNAs.** tRNAs for Val3, Ile2 and Phe were isolated by CCC[52]. The other tRNAs were isolated by RCC[24]. The isolated tRNAs and total RNA as a maker were resolved by electrophoresis on a 10% (w/v) polyacrylamide gel containing 7 M urea and 1 × TBE, and stained with SYBR Gold. Smearing of bands is due to the high GC content of tRNAs. The precursor tRNAs for Ile2 and Phe containing an intron are indicated by red triangles. 5S rRNA, Class II tRNAs, and Class I tRNAs are indicated. We confirmed the reproducibility of this result. The unprocessed gel images are provided in Supplementary Fig. 10.

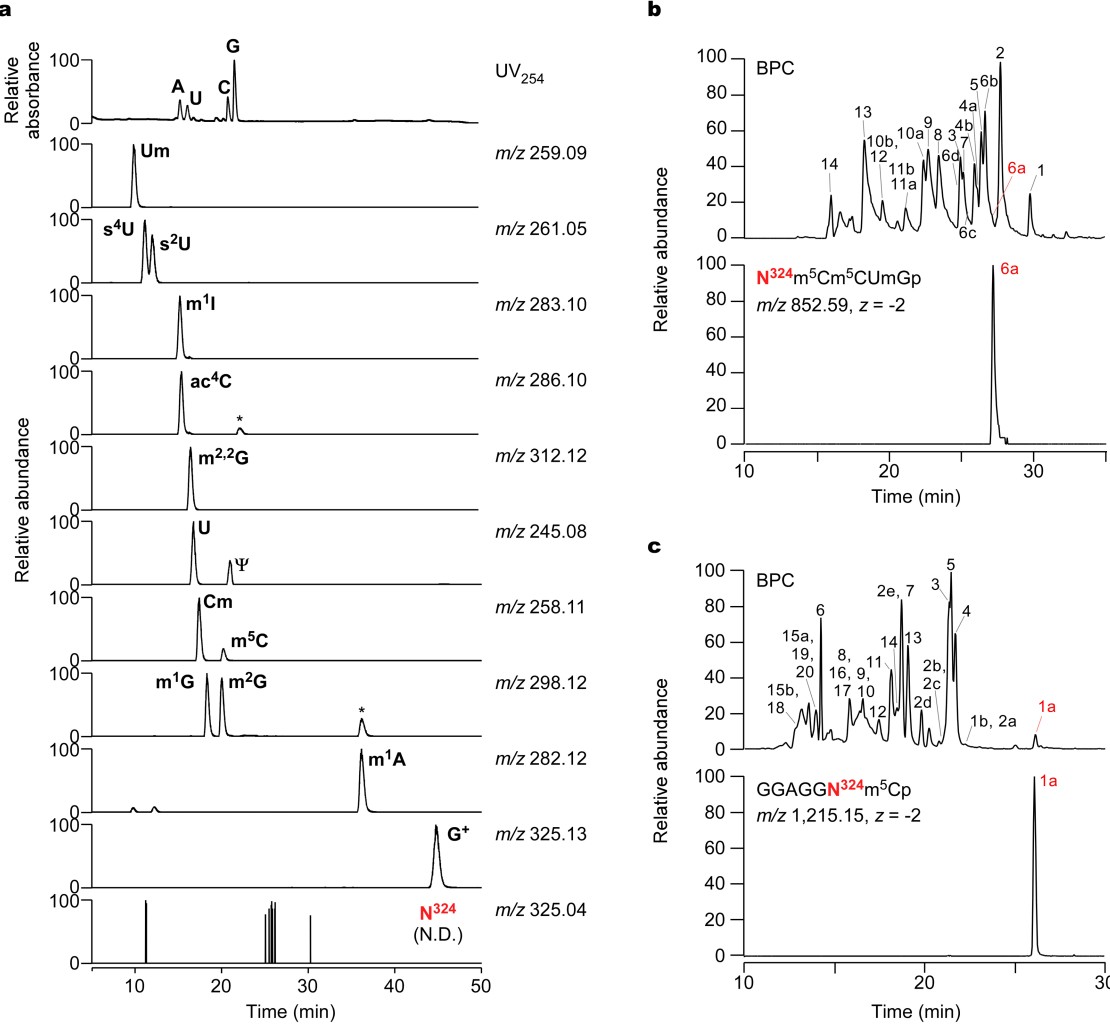

**Extended Data Fig. 2 | Nucleoside analysis and RNA-MS of _S. tokodaii_ tRNA[Val2/3]. (a)** Nucleosides of _S. tokodaii_ tRNA[Val2/3] were subjected to LC/MS analysis. Top panel shows the UV trace at 254 nm. Second to bottom panels show XICs detecting proton adducts of modified nucleosides as indicated. N[324] nucleoside was not detected in the bottom panel. m[1]G is derived from tRNA[Val2], which was co-isolated with tRNA[Val3] (Supplementary Fig. 1). Asterisks indicate unassigned ions. **(b)** RNA-MS of RNase T₁ digests of the isolated _S. tokodaii_ tRNA[Val2/3]. The upper panel shows the BPC of RNase T₁-digested

fragments. The sequence and molecular mass of each RNA fragment numbered in BPC are listed in Supplementary Table 1a. The lower panel shows the XIC of the divalent negative ion of the fragment containing N[324]. **(c)** RNA-MS of RNase A digests of isolated _S. tokodaii_ tRNA[Val2/3]. The upper panel shows the BPC of RNase A-digested fragments. The sequence and molecular mass of each RNA fragment numbered in BPC are listed in Supplementary Table 1b. The lower panel shows the XIC of the divalent negative ion of the fragment containing N[324].

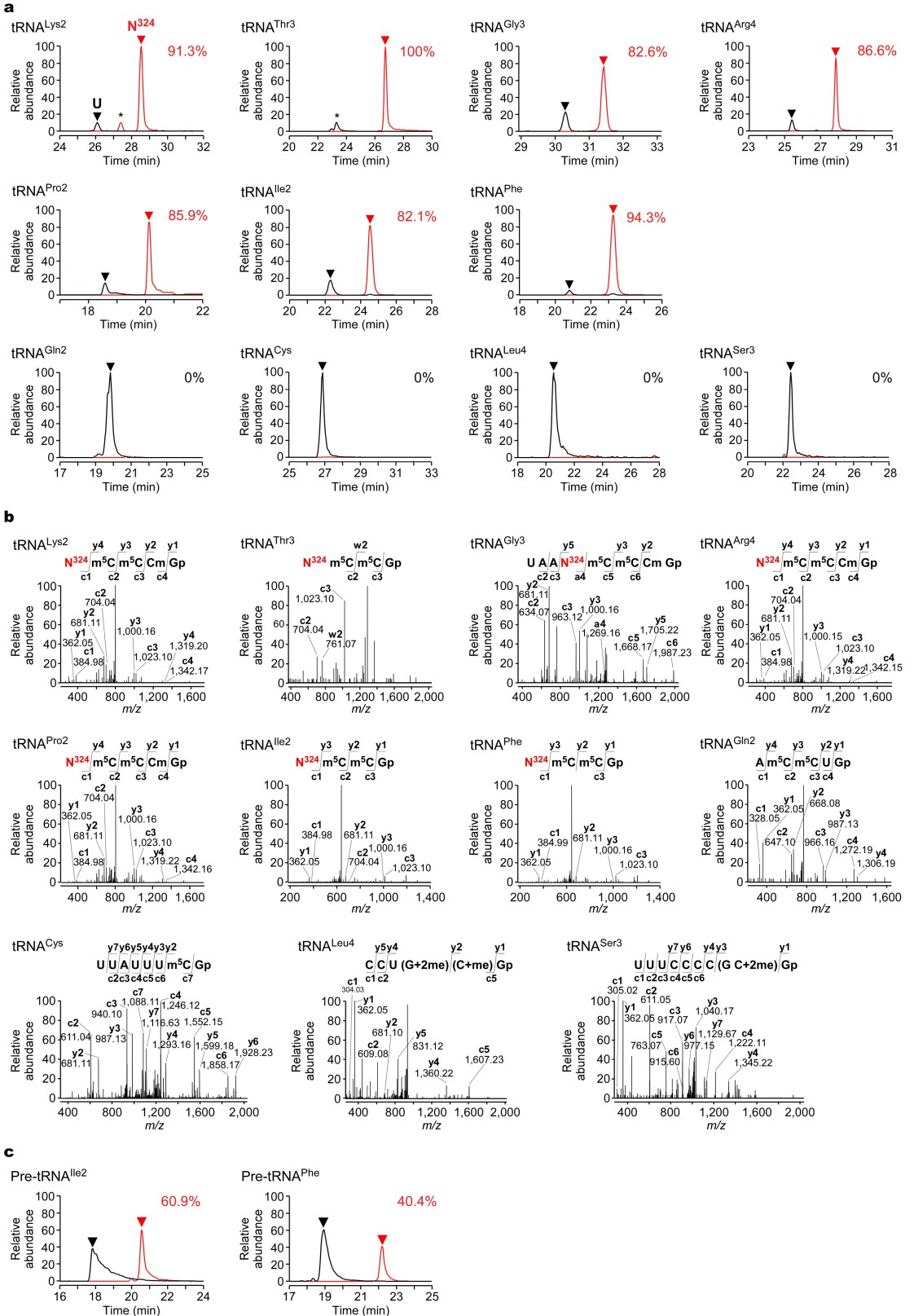

**Extended Data Fig. 3 | N^324 is present in various class I tRNAs. (a)** RNA-MS of RNase T$_1$ digests of the isolated *S. tokodaii* tRNAs. XICs show negative ions of RNA fragments derived from V-loop containing N^324 (red line) or U (black line) at position 47. Sequence and *m/z* value of each fragment are provided in Supplementary Table 2. Modification frequency of N^324 indicated in each tRNA was calculated from relative peak intensities of the modified and unmodified fragments. Unassigned fragments are indicated by asterisks. **(b)** CID spectrum of the RNA fragments detected in **(a)**. The negatively-charged ion of each fragment was used as a precursor ion for CID analysis. The product ions in the CID spectrum are assigned on the sequences. N^324 is shown in red. **(c)** N^324 is introduced in precursor tRNAs. RNA-MS of RNase T$_1$ digests of the precursor tRNAs for Ile2 and Phe isolated from *S. tokodaii* (Extended Data Fig. 1). XICs show negative ions of the RNA fragments derived from V-loop containing N^324 (red line) or U (black line) at position 47. Modification frequency of N^324 indicated in each tRNA was calculated from relative peak intensities of the modified and unmodified fragments.

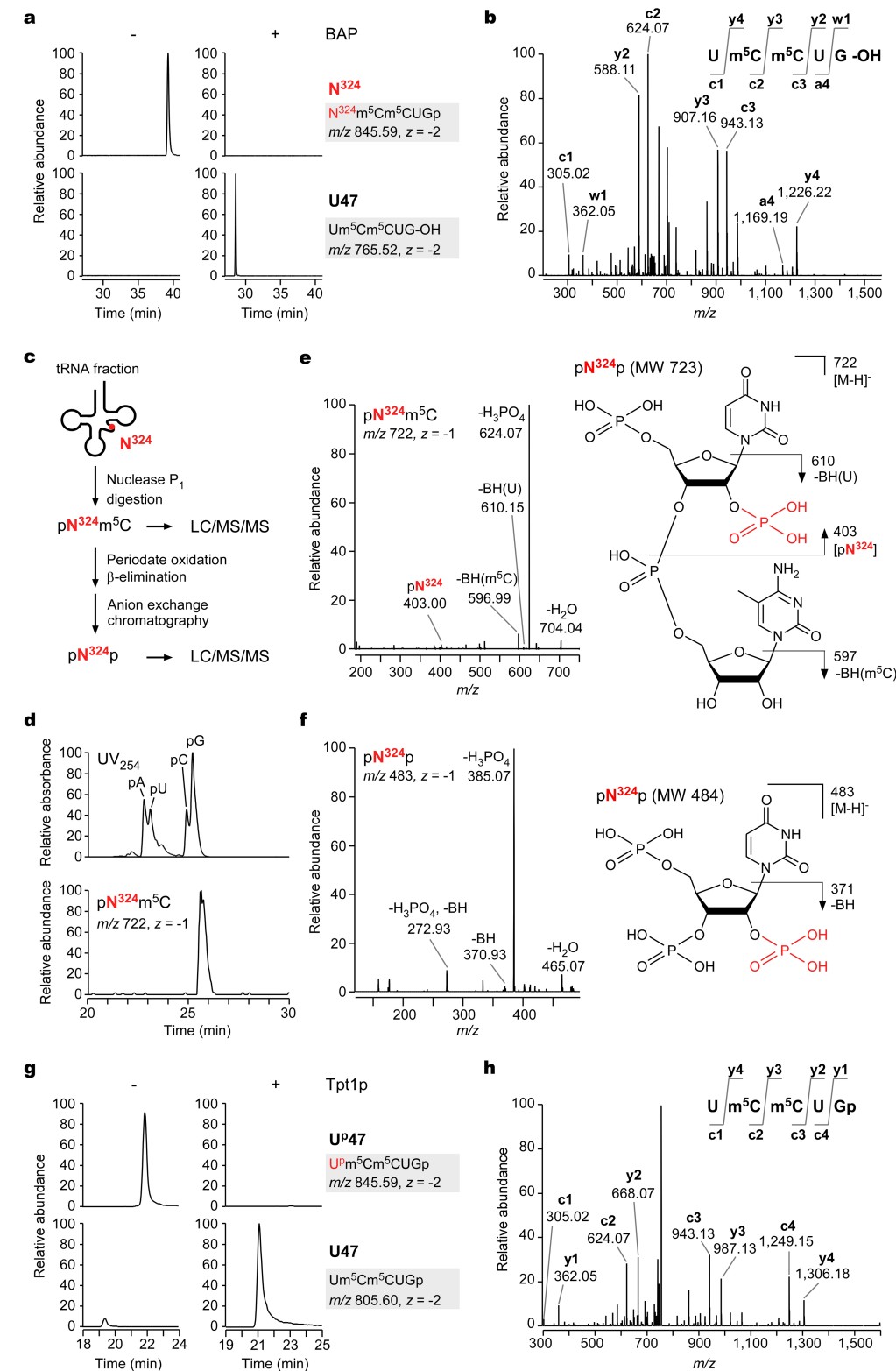

**Extended Data Fig. 4** | See next page for caption.

**Extended Data Fig. 4 | Chemical structure determination of $N^{324}$.** (**a**) RNA-MS of the $N^{324}$-containing fragment of tRNA$^{Val3}$ digested with RNase $T_1$, with (+) or without (-) BAP treatment. XICs show the divalent negative ions of $N^{324}m^5Cm^5CUGp$ (*m/z* 845.59, *z* = -2) and $Um^5Cm^5CUG$-$_{OH}$ (*m/z* 765.52, *z* = -2). Two phosphates were removed by this treatment. (**b**) CID spectrum of the $N^{324}$-containing fragment treated with BAP. The product ions are assigned on $Um^5Cm^5CUG$-$_{OH}$. (**c**) Preparation scheme of $pN^{324}p$. *S. tokodaii* tRNA fraction is digested with nuclease $P_1$, yielding dinucleotide $pN^{324}m^5C$. The digests were subjected to periodate oxidation and β-elimination to remove the 3′ terminal residue. The resultant $pN^{324}p$ was purified by anion exchange chromatography and subjected to LC/MS/MS analysis. (**d**) LC/MS nucleotide analysis of the nuclease $P_1$ digest of *S. tokodaii* tRNA fraction. UV trace at 254 nm (upper panel) and XIC of the negatively charged ion of $pN^{324}m^5C$ (*m/z* 722, *z* = -1) (lower panel) are shown. (**e**) CID spectrum of $pN^{324}m^5C$. The product ions were assigned on the predicted chemical structure of $pN^{324}m^5C$. The phosphate group of $N^{324}$ is shown in red. (**f**) CID spectrum of the $N^{324}$ nucleotide ($pN^{324}p$; *m/z* 483, *z* = -1). The product ions are assigned in the predicted chemical structure of $pN^{324}p$. (**g**) RNA-MS of the V-loop-containing RNA fragment with (+) or without (-) Tpt1p treatment before RNase $T_1$ digestion. XICs show the divalent negative ions of $U^pm^5Cm^5CUGp$ (*m/z* 845.59, *z* = -2) and $Um^5Cm^5CUGp$ (*m/z* 805.60, *z* = -2). (**h**) CID spectrum of the dephosphorylated fragment by Tpt1p. The Tpt1p-treated tRNA$^{Val3}$ was digested with RNase $T_1$ and analyzed by RNA-MS. The V-loop containing fragment was selected as a precursor for CID. The product ions are assigned on the sequence as indicated.

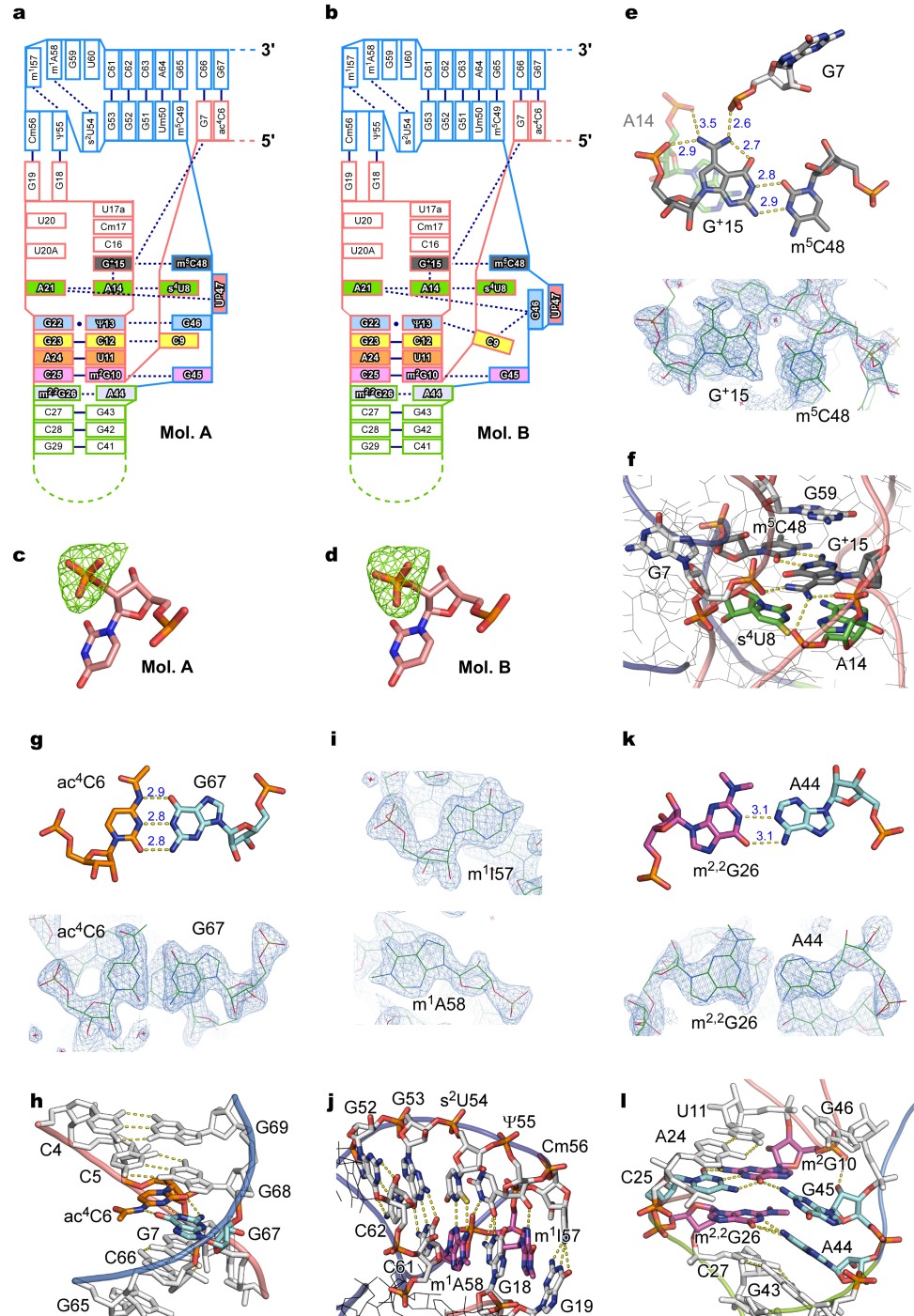

**Extended Data Fig. 5 | Crystal structures of *S. tokodaii* tRNA[Val3].**

(**a,b**) Schematic views of Mol. A (**a**) and B (**b**) of *S. tokodaii* tRNA[Val3]. Each residue is shown as a box. Color codes for the base pairs and base triples are the same as those in Fig. 2b,c. Tertiary interactions are shown as blue dashed lines. (**c,d**) Simulated annealing-omit $F_o$-$F_c$ map contoured at 3.0 sigma around 2′-phosphate of Mol. A (**c**) and Mol. B (**d**). (**e**) Levitt base pair of G[+]15 and m[5]C48 with neighboring residues shown in stick representation with electron density map. $2F_o$-$F_c$ electron density map contoured at 0.76 sigma around G[+]15–m[5]C48 is shown in the lower panel. (**f**) Close-up view of the tRNA core around the G[+]15–m[5]C48 base pair. G[+]15, m[5]C48, and neighboring residues are shown in stick representation. Other residues are indicated as lines. Backbones are shown as cartoons. Hydrogen bonds are indicated by yellow dash lines. (**g**) Base pair of

ac[4]C6 with G67 in stick representation with electron density map. $2F_o$-$F_c$ electron density map contoured at 0.76 sigma around ac[4]C6–G67 is shown in the lower panel. (**h**) Close-up view of the acceptor stem including ac[4]C6, G67, and neighboring base-pairs and nucleotides. (**i**) Electron density map of m[1]I57 and m[1]A58. $2F_o$-$F_c$ electron density maps contoured at 0.76 sigma for m[1]I57 and 1.02 sigma for m[1]A58 are shown. (**j**) Close-up view of T-loop including m[1]I57, s[2]U54, m[1]A58, and neighboring base-pairs and nucleotides. (**k**) Base pair of m[2,2]G26 with A44 in stick representation with electron density map. $2F_o$-$F_c$ electron density map contoured at 0.76 sigma around m[2,2]G26–A44 is shown in the lower panel. (**l**) Close-up view of D- and anticodon-stems including m[2]G10, m[2,2]G26, and neighboring base-pairs and nucleotides.

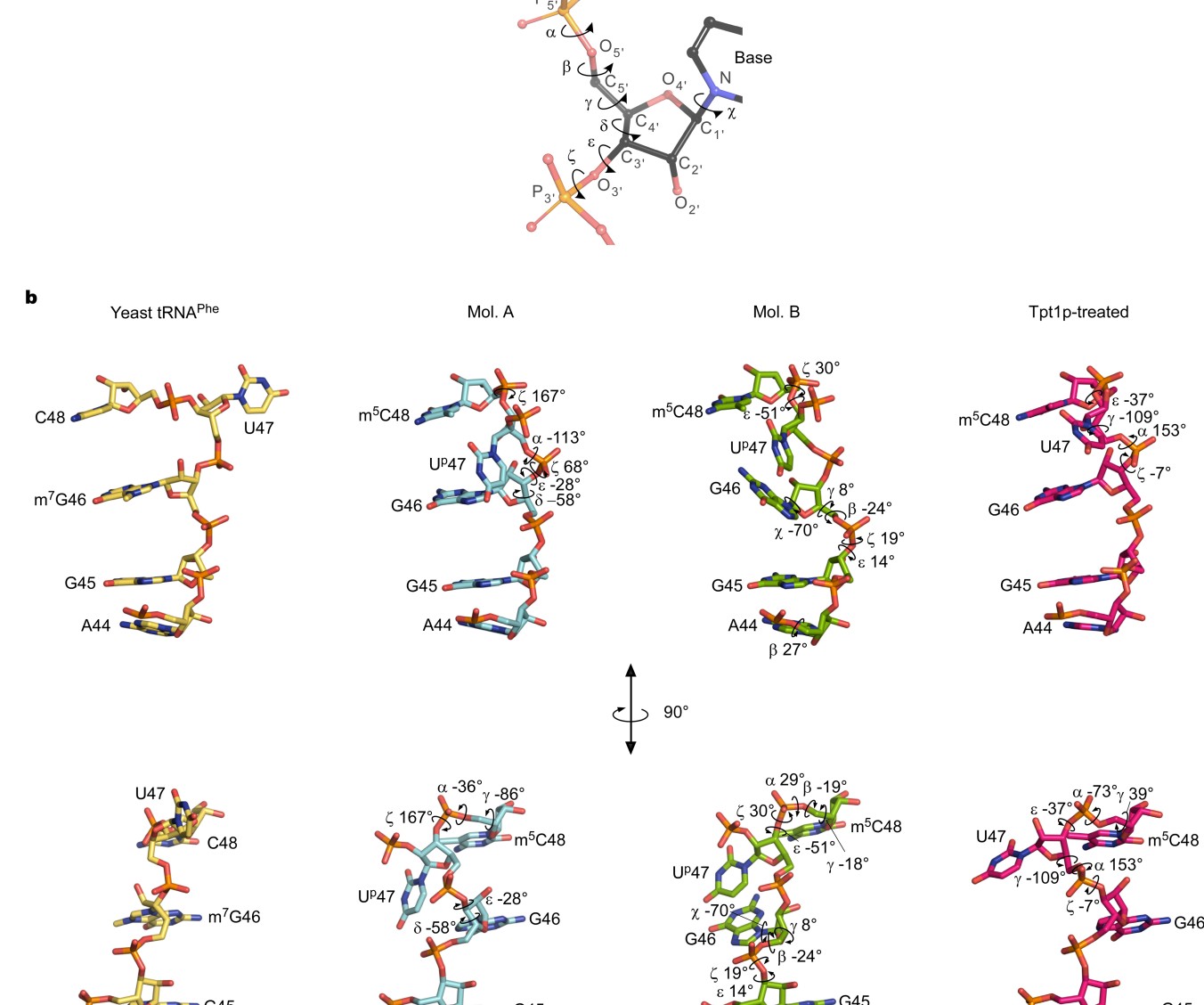

**Extended Data Fig. 6 | Torsion angles of nucleotides around position 47 of tRNAs.** (**a**) Key to torsion angles of nucleic acid backbone. Nomenclature of each angle is shown next to its direction, depicted as a black curved arrow. (**b**) Comparison of torsion angles at positions 44–48 of *S. cerevisiae* tRNA[Phe] (PDB: 1EHZ) (leftmost), and *S. tokodaii* tRNA[Val3] Mol. A (left), Mol. B (right), and Tpt1p-treated tRNA[Val3] (rightmost). Torsion angles of each tRNA are listed in

Supplementary Table 3. Torsion angle changes from yeast tRNA[Phe] to Mol. A are shown on Mol. A as curved arrows. Torsion angle changes from Mol. A to Mol. B are shown on Mol. B as curved arrows. Torsion angle changes from Mol. A to Tpt1p-treated tRNA (Mol. A) are shown on Tpt1p-treated tRNA as curved arrows. The lower panels show the 90 degree-rotated models.

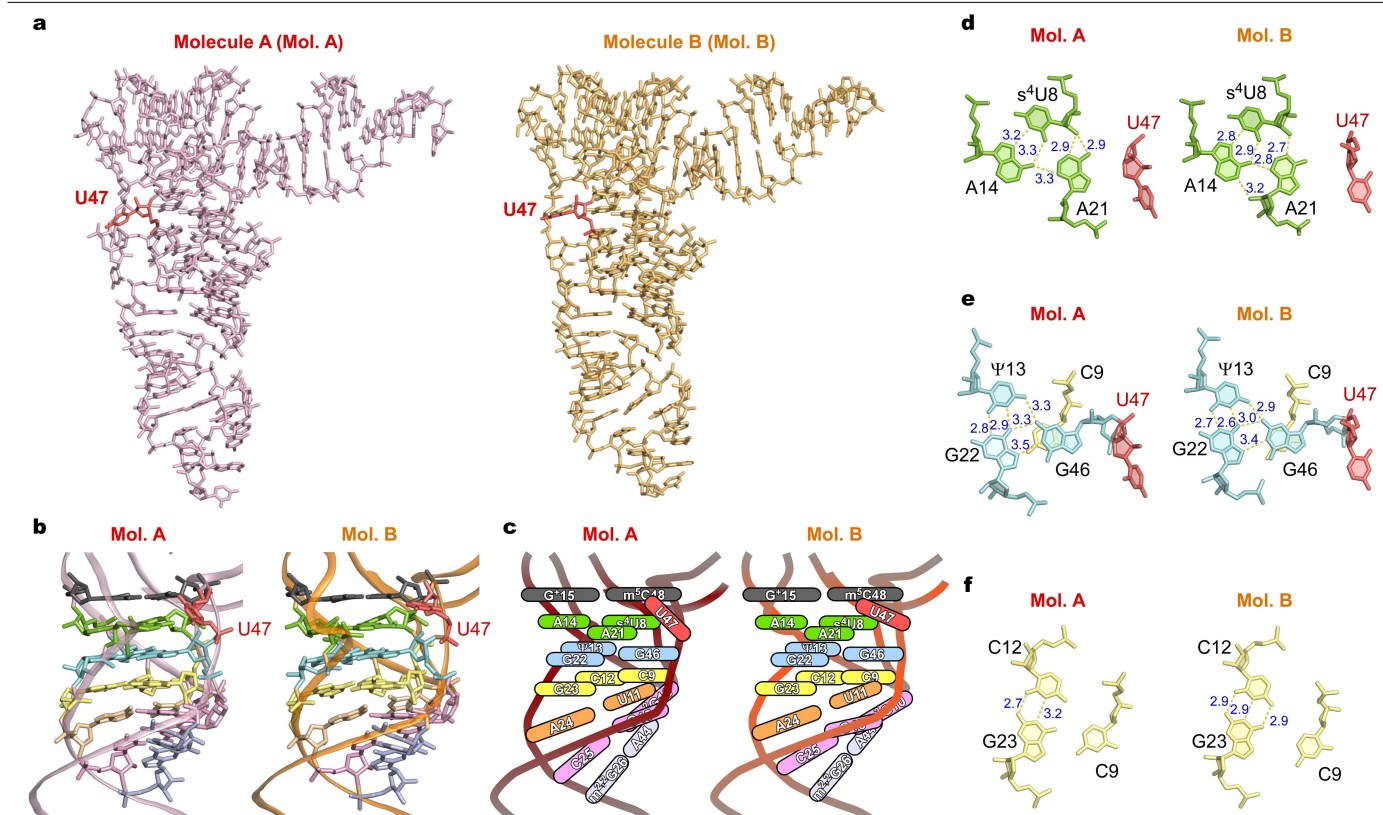

**Extended Data Fig. 7 | Crystal structure of Tpt1p-treated *S. tokodaii* tRNA^Val3.** (**a**) Overviews of crystal structure of Tpt1p-treated *S. tokodaii* tRNA^Val3 with stick representation. Molecules A (left) and B (right) are shown in stick representation in pink and orange, respectively. U47 is colored in red. (**b**) Close-up views of the core structure of Mol. A (left) and B (right). Color code is the same as in Fig. 2b. (**c**) Schematic views of the core structure of Mol. A (left) and B (right). (**d, e, f**) Atomic structures of the base triples s^4U8–A14–A21 (top), Ψ13–G22–G46 (middle) and C12–G23–C9 (bottom), in the core region of Mol. A (left) and B (right). Dashed lines indicate predicted interactions, with bond length in Å. U47 is shown in red.

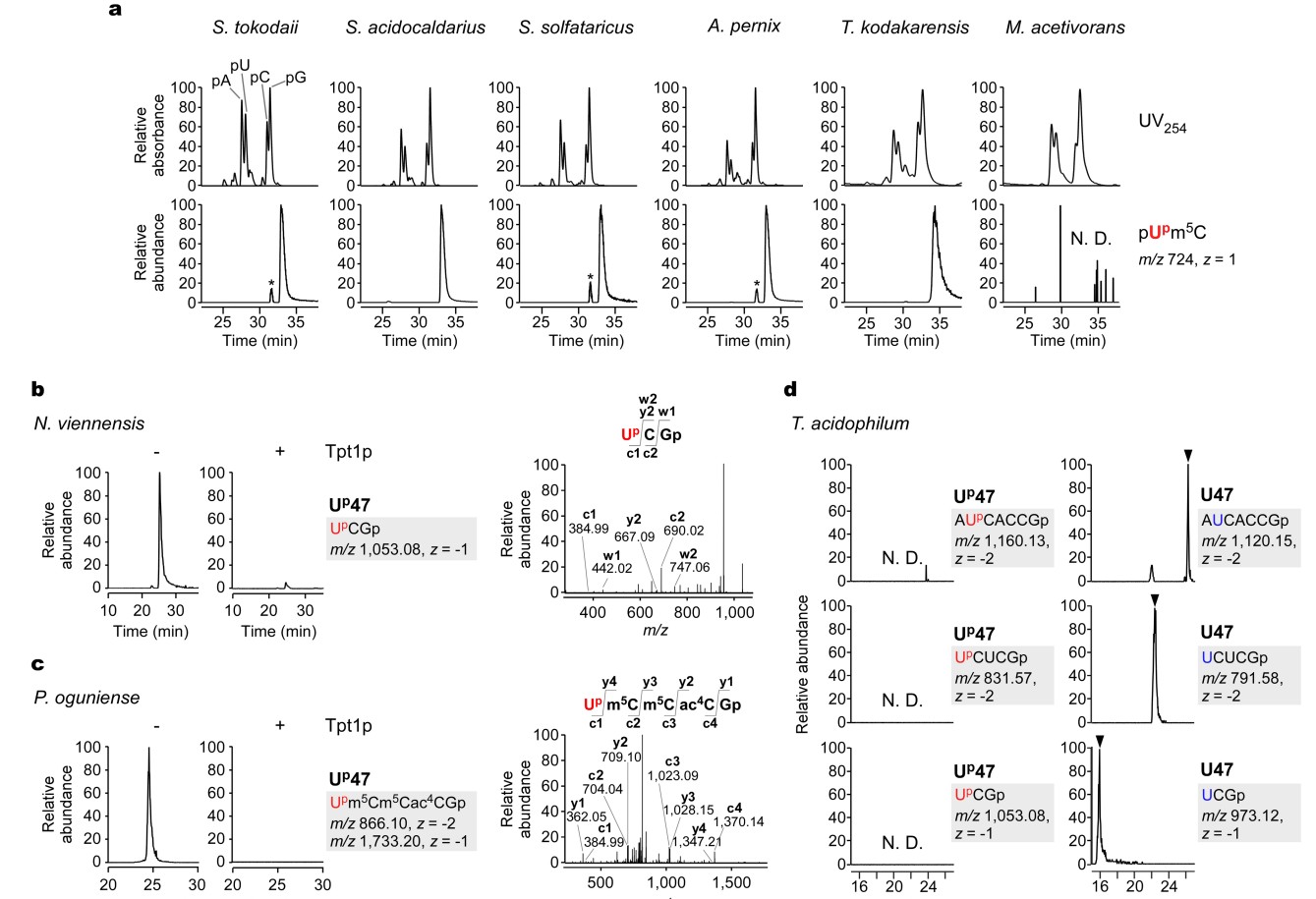

**Extended Data Fig. 8 | Phylogenetic distribution of U^p47 in archaeal species.** (**a**) LC/MS nucleotide analyses of tRNA fractions from *S. tokodaii*, *S. acidocaldarius*, *S. solfataricus*, *A. pernix*, *T. kodakarensis*, and *M. acetivorans*. UV trace at 254 nm (upper panels) and XICs of the proton adducts of pN^{324}m^5C (*m/z* 724, *z* = 1) (lower panels) are shown. Asterisks indicate unassigned ions. (**b**) RNA-MS shotgun analysis of *N. viennensis* tRNA fraction treated with (right panel) or without (left panel) Tpt1p before RNase T_1 digestion. XICs show the RNA fragments containing U^p47 (U^pCGp; *m/z* 1,053.08, *z* = -1). The product ions in the CID spectrum are assigned on U^pCGp. (**c**) RNA-MS shotgun analysis of *P. oguniense* tRNA fraction treated with (right panel) or without (left panel) Tpt1p before RNase T_1 digestion. XICs show the RNA fragments containing U^p47 (U^pm^5Cm^5Cac^4CGp; *m/z* 866.10, 1,733.20, *z* = -2, -1). The product ions in the CID spectrum are assigned on U^pm^5Cm^5Cac^4CGp. (**d**) RNA-MS shotgun analysis of *T. acidophilum* tRNA fraction digested by RNase T_1. XICs show the expected RNA fragments containing U^p47 (left panels) or U47 (right panels) as indicated.

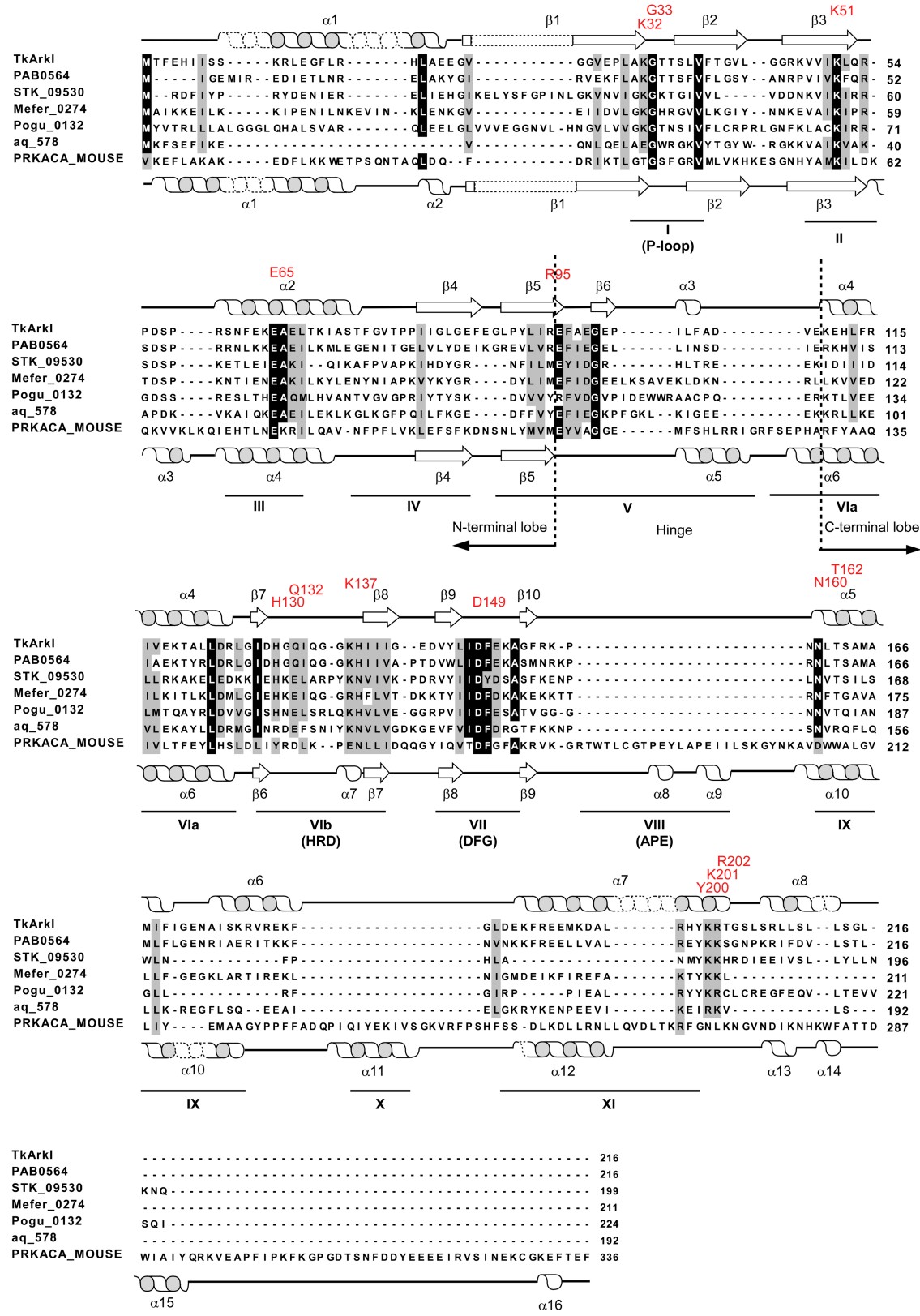

**Extended Data Fig. 9 | Sequence alignment of ArkI family (COG2112).** ArkI homologs and PRKACA (PRKACA_MOUSE, PDB: 1ATP) as a canonical ePK are aligned based on structure comparison using DALI (http://ekhidna2.biocenter. helsinki.fi/dali/). Bacterial and archaeal homologs of ArkI are added using MAFFT (https://mafft.cbrc.jp/alignment/server/). Black and gray boxes indicate the degree of sequence similarity. Residues mutated in TkArkI are indicated as red letters. The alpha helices and beta strands observed in the TkArkI structure are depicted on top of alignments as helices and arrows, respectively. Those observed in PRKACA are depicted under the alignments, as well. Subdomains (I to XI) and representative motifs (P-loop, HRD, DFG, and APE) in ePK are underlined and featured. Abbreviations for organisms: Tk, *Thermococcus kodakarensis*; PAB, *Pyrococcus abyssi*; STK, *Sulfurisphaera tokodaii*; Mefer, *Methanocaldococcus fervens*; Pogu, *Pyrobaculum oguniense*; aq, *Aquifex aeolicus*.

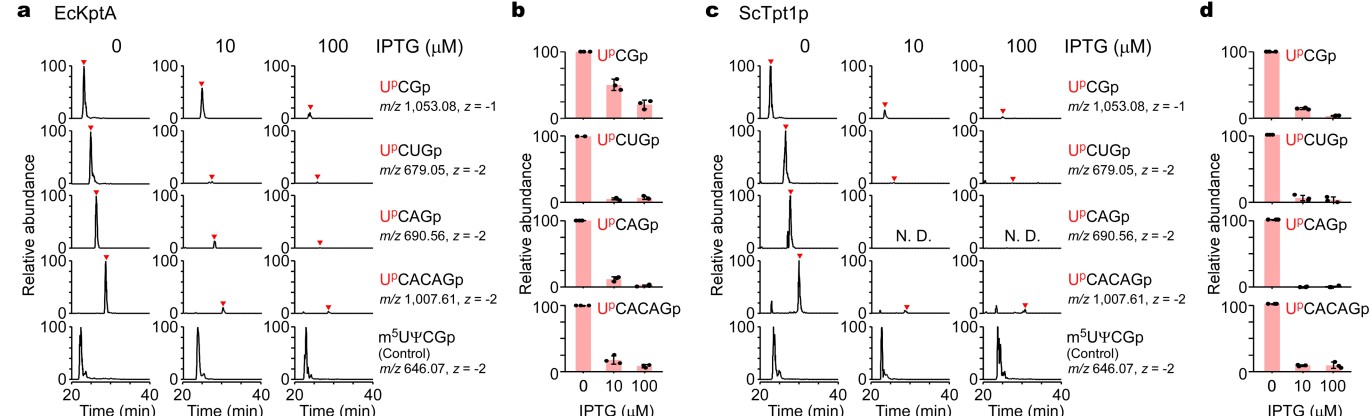

**Extended Data Fig. 10 | Dephosphorylation of U^P47 by KptA/Tpt1p homologs in *E. coli*.** (**a**, **c**) *In vivo* dephosphorylation of U^P47 by EcKptA (**a**) or ScTpt1p (**c**). XICs show U^P47-containing fragments from various *E. coli* tRNA species (Supplementary Table 5) isolated from *E. coli* Δ*trmB*/Δ*tapT* strain expressing *N. viennensis* ArkI; U^PCGp (top panels), U^PCUGp (second panels), U^PCAGp (third panels), U^PCACAGp (fourth panels), and m^5UΨCGp as a control fragment (bottom panels). Relative abundance of the U^P47-containing

fragments was measured in *E. coli* strain in which EcKptA (**a**) or ScTpt1p (**c**) is not expressed (left panels) or induced by 10 μM (middle panels) or 100 μM IPTG (right panels). (**b**, **d**) Quantification of U^P47 dephosphorylation in *E. coli* by EcKptA (**b**) or ScTpt1p (**d**). Peak intensity is shown for each U^P47-containing fragment detected in tRNA fractions from *E. coli* strain cultured with 0, 10, or 100 μM IPTG. Data represent average values of technical triplicates ± s.d.

**Extended Data Table 1 | Data collection and refinement statistics.**

| | Wild-type $St$ tRNA$^{Val}$ | Dephosphorylated $St$ tRNA$^{Val}$ | $Tk$ArkI_KI | Native $Tk$ArkI |
|---|---|---|---|---|
| **Data collection** | | | | |
| Space group | $P2_1$ | $P2_1$ | $P6_1$ | $P6_1$ |
| Cell dimensions | | | | |
| $a, b, c$ (Å) | 32.65, 116.62, 57.24 | 32.56, 116.68, 56.14 | 68.03, 68.03, 99.45 | 66.85, 66.85, 98.28 |
| $\alpha, \beta, \gamma$ (°) | 90, 100.97, 90 | 90, 101.34, 90 | 90, 90, 120 | 90, 90, 120 |
| Wavelength (Å) | 0.98 | 0.98 | 1.5 | 1.0 |
| Resolution (Å)* | 50-1.9 (1.97-1.90) | 50-2.6 (2.71-2.61) | 50-2.4 (2.48-2.40) | 50-1.8 (1.87-1.80) |
| $R_{sym}$* | 0.068 (1.552) | 0.111 (1.090) | 0.213 (2.887) | 0.069 (1.810) |
| $<I / \sigma I>$* | 15.7 (1.6) | 14.5 (2.0) | 20.5 (2.9) | 35.2 (2.7) |
| $CC_{1/2}$* | 0.998 (0.609) | 0.997 (0.649) | 0.999 (0.787) | 1.000 (0.882) |
| Completeness (%)* | 98.8 (97.5) | 99.8 (99.5) | 100.0 (99.4) | 100.0 (99.8) |
| Redundancy* | 6.7 (6.8) | 6.7 (6.2) | 39.0 (38.9) | 39.6 (40.5) |
| | | | | |
| **Phasing** | | | | |
| I sites | | | 10 | |
| FOM | | | 0.322 | |
| | | | | |
| **Refinement** | | | | |
| Resolution (Å) | 50-1.9 | 50-2.6 | | 50-1.8 |
| No. reflections | 32414 | 12511 | | 23014 |
| $R_{work}$ / $R_{free}$ (%) | 19.47 / 21.98 | 22.49 / 26.23 | | 17.10 / 20.49 |
| No. atoms | | | | |
| RNA | 3368 | 3360 | | 1701 |
| Ligand | - | - | | 20 |
| Water | 92 | 67 | | 77 |
| $B$-factors (Å$^2$) | | | | |
| RNA | 53.8 | 51.3 | | 42.18 |
| Ligand | - | - | | 53.32 |
| Water | 39.9 | 39.0 | | 47.48 |
| R.m.s. deviations | | | | |
| Bond lengths (Å) | 0.007 | 0.005 | | 0.012 |
| Bond angles (°) | 0.95 | 1.06 | | 1.11 |
| Estimated mean coordinate error | | | | |
| Phenix maximum likelihood (Å) | 0.24 | 0.43 | | 0.17 |

*Values in parentheses are for the highest-resolution shells.

# Reporting Summary

## Statistics

For all statistical analyses, confirm that the following items are present in the figure legend, table legend, main text, or Methods section.

| n/a | Confirmed | |
|---|---|---|
| ☐ | ☒ | The exact sample size ($n$) for each experimental group/condition, given as a discrete number and unit of measurement |
| ☐ | ☒ | A statement on whether measurements were taken from distinct samples or whether the same sample was measured repeatedly |
| ☐ | ☒ | The statistical test(s) used AND whether they are one- or two-sided<br>*Only common tests should be described solely by name; describe more complex techniques in the Methods section.* |
| ☒ | ☐ | A description of all covariates tested |
| ☒ | ☐ | A description of any assumptions or corrections, such as tests of normality and adjustment for multiple comparisons |
| ☐ | ☒ | A full description of the statistical parameters including central tendency (e.g. means) or other basic estimates (e.g. regression coefficient) AND variation (e.g. standard deviation) or associated estimates of uncertainty (e.g. confidence intervals) |
| ☐ | ☒ | For null hypothesis testing, the test statistic (e.g. $F$, $t$, $r$) with confidence intervals, effect sizes, degrees of freedom and $P$ value noted<br>*Give P values as exact values whenever suitable.* |
| ☒ | ☐ | For Bayesian analysis, information on the choice of priors and Markov chain Monte Carlo settings |
| ☒ | ☐ | For hierarchical and complex designs, identification of the appropriate level for tests and full reporting of outcomes |
| ☒ | ☐ | Estimates of effect sizes (e.g. Cohen's $d$, Pearson's $r$), indicating how they were calculated |

*Our web collection on statistics for biologists contains articles on many of the points above.*

## Software and code

Policy information about availability of computer code

| Data collection | Gel images were obtained by FLA-7000 and FLA-9000.<br>Melting curves were obtained by V-630.<br>MS data were obtained by LTQ Orbitrap XL, Q Exactive Hybrid Quadrupole-Orbitrap Mass Spectrometer, LCQ-Advantage Ion-trap Mass Spectrometer.<br>Radioactivities were measured by Tri-Carb 2910TR1<br>Cell densities were monitored by S1200 diode array spectrophotometer.<br>X-ray diffraction data were obtained by Dectris Eiger X16MS detector and UGUA control system with BL17A beamline at the Photon Factory. |
|---|---|

| Data analysis | Canvas X (version 20), ACD/ChemSketch (Freeware, 2018.2.1) , ChemDraw (20.1.1), Excel (2016, 2019) and R (4.1.2) were used to draw figures.<br>Xcalibur (4.1) was used for mass spec analysis.<br>Spectra Manager (v2) was used for Tm measurement.<br>Multi Gauge (V3.0) was used for graphical analysis.<br>Prism 7 was used for kinetic analysis.<br>XDS/XSCALE (VERSION Feb 5, 2021) , SHELX (2016/1), Phaser (2.8.3), RESOLVE (2.15), APBS (1.5), Coot (0.8.9.1),and Phenix (Version 1.18.2-3874) were used for<br>structural analyses.<br>Coot (0.8.9.1), Pymol (2.4.0), Cuemol (2.2.3.443) were used for draw structure data.<br>RECOG (1.1.32) was used for comparative genome.<br>x3dna-dssr (v1.9.10) was used for analysis of torsion angles.<br>PhyloT (v2) and iTOL (6.5) were used for analysis of phylogenetic distribution.<br>DALI (v.5) and MAFFT (Version 7) were used for sequence alignment. |
|---|---|

For manuscripts utilizing custom algorithms or software that are central to the research but not yet described in published literature, software must be made available to editors and reviewers. We strongly encourage code deposition in a community repository (e.g. GitHub). See the Nature Portfolio guidelines for submitting code & software for further information.

## Data

Policy information about availability of data

All manuscripts must include a data availability statement. This statement should provide the following information, where applicable:

- Accession codes, unique identifiers, or web links for publicly available datasets
- A description of any restrictions on data availability
- For clinical datasets or third party data, please ensure that the statement adheres to our policy

Public databases: Microbial Genome Database(MBGD), NCBI database, COG database, BacDive, Genome Online Database, IMG database, Mongo Oligo Mass Calculator v2.08, Genomic tRNA database, Modomics, Protein Data Bank (PDB) (1EHZ, 1IVS, and 1ATP). Coordinates and structure factors have been deposited in PDB under accession code 7VNV, 7VNW, and 7VNX.

# Field-specific reporting

Please select the one below that is the best fit for your research. If you are not sure, read the appropriate sections before making your selection.

☒ Life sciences          ☐ Behavioural & social sciences          ☐ Ecological, evolutionary & environmental sciences

For a reference copy of the document with all sections, see nature.com/documents/nr-reporting-summary-flat.pdf

# Life sciences study design

All studies must disclose on these points even when the disclosure is negative.

| Sample size | 3 data points were used in student t-test. |
|---|---|
| Data exclusions | No data were excluded. |
| Replication | All attempts to replicate experiments succeeded.<br>Tm measurement, RNase probing, growth comparison, in vitro biochemical studies, were technically or biologically triplicated. |
| Randomization | Randomization was irrelevant for this basic study because it did not involve clinical trials or population studies. |
| Blinding | No blinding was required because the results of measurement or analysis was not affected by knowledge of sample identities. |

# Reporting for specific materials, systems and methods

We require information from authors about some types of materials, experimental systems and methods used in many studies. Here, indicate whether each material, system or method listed is relevant to your study. If you are not sure if a list item applies to your research, read the appropriate section before selecting a response.

## Materials & experimental systems

| n/a | Involved in the study |
|-----|----------------------|
| ☒ ☐ | Antibodies |
| ☒ ☐ | Eukaryotic cell lines |
| ☒ ☐ | Palaeontology and archaeology |
| ☒ ☐ | Animals and other organisms |
| ☒ ☐ | Human research participants |
| ☒ ☐ | Clinical data |
| ☒ ☐ | Dual use research of concern |

## Methods

| n/a | Involved in the study |
|-----|----------------------|
| ☒ ☐ | ChIP-seq |
| ☒ ☐ | Flow cytometry |
| ☒ ☐ | MRI-based neuroimaging |

nature portfolio | reporting summary

March 2021