## [Peer Review File · Nature]

Manuscript Title: Reversible RNA phosphorylation stabilizes tRNA for cellular thermotolerance

Reviewer Comments & Author Rebuttals

Reviewer Reports on the Initial Version:

Referee #1:

In this technical tour de force, Ohira et al. demonstrate for the first time the internal phosphorylation of tRNA (in several hyperthermophilic archaea), determine the structure of the phosphorylated and dephosphorylated tRNAs using material purified from source, identify and determine the structure of the responsible kinase, identify the corresponding phosphatase, demonstrate biophysically that phosphorylation leads to increased thermostability of the tRNA in vitro, and by deletion of the kinase, that the modification is important for thermotolerance in live cells (and this is not a comprehensive list of firsts). While tRNA modifications (and RNA modifications, in general, as the "epitranscriptome") have attracted considerable attention, the link between a modification's biochemical role and physiological function is usually very elusive. Not here. In a single manuscript, Ohira et al. have achieved a comprehensive characterization, ranging from biochemistry to structure to cell physiology of a modification they have discovered. In so doing, they have also opened the door to future studies that characterize the signal transduction that controls reversible phosphorylation. The responsible kinase is also present in thermophilic bacteria, and surely homologs will be discovered that carry out (possibly for other adaptive reasons) similar reactions in eukaryotes.

The significance of the work is extremely high, as tRNA is the most abundant RNA in cells, is essential for cell function, and adaptive stabilization (thermally in this case) of RNA is paradigm-shifting observation that will surely apply to other cellular RNAs. Moreover, the discovery of a class of RNA-directed kinases related to protein kinases raises the question of whether protein kinases did not evolve from ancestral RNA kinases; if so, a whole universe of RNA epitranscriptomics may await.

I think the work, independent of its technical brilliance and novel avenues it opens, will be broadly interesting to biochemists, molecular biologists, signaling specialists and microbiologists, and is eminently suitable for publication in this venue.

I have only a handful of minor suggestions regarding presentation.

1. Regarding the two conformations observed in the fully modified tRNA crystal structure asymmetric unit: Since the crystals are grown in non-thermophilic conditions, it remains but a speculation that the non-canonical core is on-pathway for melting. This is certainly a possibility the

authors can entertain, but I think it's certainly not "implied" by their analysis (top of p. 9). Equally plausible is that the modification confers on the tRNA the ability to adopt multiple near-iso-energetic folds, thus increasing the entropy of the folded ensemble, and in this way increasing thermostability. Or the mol B conformation is an artifact of crystallizing the tRNA at 20 °C, i.e. it is a cold-denatured form of the RNA. These are some other possibilities, which may well be incorrect, but my point is that the assumption that Mol B conformation is on-pathway is not proven (or even analyzed) at this point.

2. Fig. 3a inset: The electron density shown should be a simulated annealing-omit Fo-Fc map. Moreover, since the dataset is quite complete, it is likely that if the authors were to scale their data with the Friedel pairs separate and compute an anomalous difference Fourier synthesis, they would observe the anomalous signal at the phosphorus in question (as well as the other phosphoruses in the structure), thus providing a crystallographic correlate to their MS analyses.

3. Table S3: Please include here (or in methods) an estimate of mean coordinate precision (e.g., Luzzati, sigmaA, Phenix maximum likelihood). Also, it is the mean intensity divided by the mean sigma of the intensity, $\langle I \rangle / \langle s(I) \rangle$.

Referee #2:

This paper from the Tomita/Suzuki laboratories presents a novel finding that 2'-phosphouridine (Up) exists at position 47 of several tRNAs from thermophilic archaea. Importantly, the authors show that this modification is under enzyme-mediated reversible regulation to control tRNA stability for thermal adaption. The authors determined the structure of *S. tokodaii* tRNAVal3 at 1.9 Å (with and without the Up). This structure, together with the biochemistry evidence, suggests that Up is indispensable for stabilizing tRNAVal. To further investigate the biological significance of this new tRNA modification, the authors identified both the kinase and the phosphatase enzymes of this tRNA internal phosphorylation. They determined the structure of the kinase, ArkI, and provided an important structural analysis of the catalytic residues and putative RNA binding interface of ArkI. The authors also revealed the phosphatase of Up47 in tRNA for reversible regulation. Lastly, the authors provided some in vivo work showing that the arkI KO and arkI + queE DKO strains present weak growth sensitivity, indicating Up and concerted action of two tRNA mods are biologically significant in thermal adaptation to some extent.

The work is done thoughtfully and with a sense of rigor. A combination of analytical chemistry, structural biology, biochemistry, and some in vivo work collectively provided a comprehension of the role of a novel tRNA internal modification. This work is a continuous success from the Tomita/Suzuki laboratories to systematically reveal a full picture of the function of enzyme-mediated tRNA modifications. I believe the paper is meritorious for publication in Nature, assuming a few issues outlined below can be addressed.

Specific Comments:

1. The authors updated their tRNA purification system and successfully purified tRNAs from a

thermophilic organism. They leveraged their expertise using Orbitrap and LC-MS/MS and identified the new internal phosphorylation site in tRNA. It is a nice presentation and explanation of how they assigned the observed phosphorylation to the 2' position in type I tRNAs, and why the traditional LC-MS/MS missed this modification. Interestingly, 84.5% N324 (Up) occurs with non-modified U50, and 12.3% occurs with modified Um50. Is there any correlation between these two modified sites (U47 and U50)?

2. The authors identified Up in tRNA precursors (tRNA^{Ala}2 and tRNA^{Phe}), and the levels of Up increased during tRNA maturation. Can the authors comment on why the Up level increased along with tRNA maturation? (Is it because Up stabilizes the modified tRNAs, while the unmodified decays along with maturation?)

3. The authors determined the structure of tRNA^{Val}3 at the Up-modified and -unmodified (Tpt1p-treatment) forms at high resolution. At the Up-modified state, the structure contains two molecules (A and B) per unit. When mol A and mol B are both Up-modified, they are in two different conformations: mol A forms a canonical tRNA core, and B has an altered core. The major change is that G46 dissociates from the base-triple Ψ 13–G22–G46 and stacks with Up47 in mol B. These structural changes are considered induced by Up47 to stabilize the tRNA structures.

(1) The authors were very rigorous and provided a structure with Tpt1p-treatment (unmodified U), and the unmodified tRNA presents a canonical core structure as defined in mol A but not mol B in the Up-modified structure. Do the authors mean even the tRNA is Up-modified, the tRNA can adopt different structures, and only the B conformation stabilizes the structure? (This might explain the small differences in Fig. 2a, although Fig. 2b convinces this reviewer that Up47 stabilizes tRNA in vitro.)

(2) The electron density for Up47 is very clear. However, could the authors provide Fo – Fc to better present the modification?

(3) (minor) Could the authors please change the color in Fig. 3e? It is hard (for this reviewer and maybe other readers) to distinguish the white and light cyan as indicated.

4. It is exciting to see the authors provide solid bioinformatics, biochemistry, and structural evidence (1.8 Å resolution for ArkI) that ArkI and KptA are the kinase and the phosphatase.

(1) Could the authors comment on whether ArkI specifically modifies tRNA but no other RNAs? Some cursory analysis or comments on other natural RNA targets would have been helpful.

(2) How does ArkI specifically recognize tRNA? Does ArkI harbor a structure-specific, tRNA-binding motif? This motif generally recognizes the outside corner of the L-shaped tRNA structure, which is the signature of a tRNA molecule. For example, as the Arc1p domain of yeast MetRS, the EMAPII domain of human TyrRS, and the freestanding Trpb111 of the *A. aeolicus* thermophile. Some analysis of the sequence of ArkI that aligns with those orthologs would be helpful.

(3) Based on Fig. 4c, is ArkI the only tRNA kinase in this thermophilic species?

5. The biochemistry evidence is clear that ATP is the phosphate donor. However, could the authors comment on:

(1) Whether tRNA binding increased ATP binding affinity (especially ATP cannot be seen in the structure of enzyme alone)?

(2) What the role of GTP identified in the structures is? Dose GTP have a higher binding affinity than ATP, so that GTP stays in the structure but not for catalysis? Is the GTP binding site the same as the ATP binding site (might be out of the scope of the study)?

6. The structure of Arkl is *T. kodakarensis* Arkl; could the authors comment on why they express Arkl of *N. viennensis* in *E. coli* in Fig. 6?

Referee #3:

In this paper, the Suzuki lab identifies phosphorylation of the 2'OH group of a uridine in a archaeal thermophilic tRNA as a new post-transcriptional modification vent. They also identify two relevant enzymes, i.e. a kinase as a "writer" of the modification, and a phosphatase as an eraser. A crystal structure of the writer forms a meaningful basis for biochemical characterization of a non-canonical kinase motif. They furthermore report crystal structures of tRNA in phosphorylated and unphosphorylated states, and characterize their biophysical differences.

The content of this paper is clearly suitable for publication in Nature, actually several times over. The identification of phosphorylation as a new modification, including writer and eraser, is spectacular. The paper is furthermore a treasure trove of RNA modification structures within the tRNA crystal structures, which in turn, being of archaeal origin, are to my knowledge unique by themselves.

Overall, I very emphatically recommend publication.

Minor comments:

1. The number of modifications is higher than 150, and there is an update of Modomics (citation #5), so the authors might do a qualified count within that database.

2. I very much appreciate reading this as a full paper, not yet compressed into the uncomfortably dense style of the journal. I did notice, though, that the bottom half of p. 4 is essentially identical to the abstract, which makes it redundant.

3. The narration occasionally meanders between present and past tense. I suggest adhering to past tense.

4. On p. 6, the following statement is insufficiently underpinned by data, especially since the phosphorylation appears to be reversible: "Thus, N324 is a modification introduced in the precursor form, and its levels would increase during tRNA maturation."

5. On p. 7, rephrase the statement “Compared with the tRNA with Up 47, the hypomodified tRNA degraded within 5 min” more precisely.

Author Rebuttals to Initial Comments:

First, we thank the reviewers for their careful and helpful review of our manuscript, and for providing valuable suggestions for its improvement. Our point-by-point responses to each of the reviewers' comments are shown below. Changes to the main text are marked in yellow.

Response	to	Reviewer	#1's	comments
----------	----	----------	------	----------

In this technical tour de force, Ohira et al. demonstrate for the first time the internal phosphorylation of tRNA (in several hyperthermophilic archaea), determine the structure of the phosphorylated and dephosphorylated tRNAs using material purified from source, identify and determine the structure of the responsible kinase, identify the corresponding phosphatase, demonstrate biophysically that phosphorylation leads to increased thermostability of the tRNA in vitro, and by deletion of the kinase, that the modification is important for thermotolerance in live cells (and this is not a comprehensive list of firsts). While tRNA modifications (and RNA modifications, in general, as the "epitranscriptome") have attracted considerable attention, the link between a modification's biochemical role and physiological function is usually very elusive. Not here. In a single manuscript, Ohira et al. have achieved a comprehensive characterization, ranging from biochemistry to structure to cell physiology of a modification they have discovered. In so doing, they have also opened the door to future studies that characterize the signal transduction that controls reversible phosphorylation. The responsible kinase is also present in thermophilic bacteria, and surely homologs will be discovered that carry out (possibly for other adaptive reasons) similar reactions in eukaryotes.

The significance of the work is extremely high, as tRNA is the most abundant RNA in cells, is essential for cell function, and adaptive stabilization (thermally in this case) of RNA is paradigm-shifting observation that will surely apply to other cellular RNAs. Moreover, the discovery of a class of RNA-directed kinases related to protein kinases raises the question of whether protein kinases did not evolve from ancestral RNA kinases; if so, a whole universe of RNA epitranscriptomics may await.

I think the work, independent of its technical brilliance and novel avenues it opens, will be broadly interesting to biochemists, molecular biologists, signaling specialists and microbiologists, and is eminently suitable for publication in this venue.

Response: We really appreciate these positive words that encourage us.

I have only a handful of minor suggestions regarding presentation.

1. Regarding the two conformations observed in the fully modified tRNA crystal structure asymmetric unit: Since the crystals are grown in non-thermophilic conditions, it remains but a speculation that the non-canonical core is on-pathway for melting. This is certainly a possibility the authors can entertain, but I think it's certainly not "implied" by their analysis (top of p. 9). Equally plausible is that the modification confers on the tRNA the ability to adopt multiple near-iso-energetic folds, thus increasing the entropy of the folded ensemble, and in this way increasing thermostability. Or the mol B conformation is an artifact of crystallizing the tRNA at 20 °C; i.e, it is a cold-denatured form of the RNA. These are some other possibilities, which may well be incorrect, but my point is that the assumption that Mol B conformation is on-pathway is not proven (or even analyzed) at this point.

Response: As pointed out, the crystals are grown at 20°C. So, Mol. B bearing non-canonical

core might be formed in such non-physiological condition. However, it is not observed in the crystal structure of the Tpt1p-treated tRNA grown at the same condition, strongly suggesting that the Mol. B structure is induced by UP47. As suggested by this reviewer, we also agree with the idea that the Mol. B structure has near-iso-energetic fold with the canonical core. We call this fold as “metastable core structure” in the text. We don’t know whether the non-canonical core is on-pathway for melting. This is just a speculation, but it is quite natural to consider that tRNA goes back and forth between Mol. A and Mol. B at high growth temperature. In this revision, we carefully revised this section and rephrased the speculation about the Mol B conformation.

2. Fig. 3a inset: The electron density shown should be a simulated annealing-omit Fo-Fc map.

Response: We provided the simulated annealing-omit Fo-Fc map for UP47 in Mol. A (Extended Data Fig. 5c) and Mol. B (Extended Data Fig. 5d).

Moreover, since the dataset is quite complete, it is likely that if the authors were to scale their data with the Friedel pairs separate and compute an anomalous difference Fourier synthesis, they would observe the anomalous signal at the phosphorus in question (as well as the other phosphoruses in the structure), thus providing a crystallographic correlate to their MS analyses.

Response: Based on the suggestion, we examined the anomalous difference map but we cannot observe any probable peaks on the phosphorus atoms. The wavelength would not be suitable for the detection of the anomalous signal ($\lambda = 0.98$ angstrom, $f'' = 0.18$ for P). We agree that the anomalous signal would help, but even without anomalous maps, we are confident enough that the modification is phosphorylation through a series of biochemical experiments.

3. Table S3: Please include here (or in methods) an estimate of mean coordinate precision (e.g., Luzzati, sigmaA, Phenix maximum likelihood). Also, it is the mean intensity divided by the mean sigma of the intensity, $\langle I \rangle / \langle \sigma(I) \rangle$.

Response: We included the estimated mean coordinate error based on Phenix maximum likelihood to Extended Data Table 1.

Response	to	Reviewer	#2's	comments
----------	----	----------	------	----------

This paper from the Tomita/Suzuki laboratories presents a novel finding that 2'-phosphouridine (Up) exists at position 47 of several tRNAs from thermophilic archaea. Importantly, the authors show that this modification is under enzyme-mediated reversible regulation to control tRNA stability for thermal adaption. The authors determined the structure of *S. tokodaii* tRNAVal3 at 1.9 Å (with and without the Up). This structure, together with the biochemistry evidence, suggests that Up is indispensable for stabilizing tRNAVal. To further investigate the biological significance of this new tRNA modification, the authors identified both the kinase and the phosphatase enzymes of this tRNA internal phosphorylation. They determined the structure of the kinase, ArkI, and provided an important structural analysis of the catalytic residues and putative RNA binding interface of ArkI. The authors also revealed the phosphatase of Up47 in tRNA for reversible regulation. Lastly, the authors provided some in vivo work showing that the arkI KO and arkI + queE DKO strains present weak growth sensitivity, indicating Up and concerted action of two tRNA mods are biologically significant in thermal adaptation to some extent.

The work is done thoughtfully and with a sense of rigor. A combination of analytical chemistry, structural biology, biochemistry, and some *in vivo* work collectively provided a comprehension of the role of a novel tRNA internal modification. This work is a continuous success from the Tomita/Suzuki laboratories to systematically reveal a full picture of the function of enzyme-mediated tRNA modifications. I believe the paper is meritorious for publication in Nature, assuming a few issues outlined below can be addressed.

Response: We really appreciate these positive words.

Specific

Comments:

1. The authors updated their tRNA purification system and successfully purified tRNAs from a thermophilic organism. They leveraged their expertise using Orbitrap and LC-MS/MS and identified the new internal phosphorylation site in tRNA. It is a nice presentation and explanation of how they assigned the observed phosphorylation to the 2' position in type I tRNAs, and why the traditional LC-MS/MS missed this modification. Interestingly, 84.5% N324 (Up) occurs with non-modified U50, and 12.3% occurs with modified Um50. Is there any correlation between these two modified sites (U47 and U50)?

Response: In Figure 1d, we detected four RNA fragments of tRNA^{Val3} with different modification status. In the presence of U^P47, the ratio of Um50-containing fragment (12.3%) to U50-containing fragment (84.5%) is 1 : 6.9, whereas in the absence of U^P47, Um50-containing fragment (0.5%) to U50-containing fragment (2.7%) is 1 : 5.4. Thus, there might be little correlation between these two modifications.

2. The authors identified Up in tRNA precursors (tRNA^{Ile2} and tRNA^{phe}), and the levels of Up increased during tRNA maturation. Can the authors comment on why the Up level increased along with tRNA maturation? (Is it because Up stabilizes the modified tRNAs, while the unmodified decays along with maturation?)

Response: According to our *in vitro* U^P47 formation, unmodified tRNA is not a good substrate for ArkI, compared to the modified tRNAs isolated from $\Delta arkI$ cells. This indicates that other tRNA modifications contribute to efficient U^P47 formation at high temperature. Because pre-tRNAs contain hypomodified nascent transcripts, it is considered that ArkI does not efficiently introduce U^P47. In fact, U^P47 frequency of pre-tRNAs is much lower than that of mature tRNAs (Extended Data Fig. 3c). Otherwise, extra sequences at both termini and intron of pre-tRNA might interfere with U^P47 formation. Or, some factors for pre-tRNA processing might block recognition by ArkI. Moreover, as suggested by this reviewer, unmodified tRNAs might be degraded rapidly by a quality control system. We added these possibilities in Supplementary Note 1.

3. The authors determined the structure of tRNA^{Val3} at the Up-modified and -unmodified (Tpt1p-treatment) forms at high resolution. At the Up-modified state, the structure contains two molecules (A and B) per unit. When mol A and mol B are both Up-modified, they are in two different conformations: mol A forms a canonical tRNA core, and B has an altered core. The major change is that G46 dissociates from the base-triple $\Psi 13$ –G22–G46 and stacks with Up47 in mol B. These structural changes are considered induced by Up47 to stabilize the tRNA structures.

(1) The authors were very rigorous and provided a structure with Tpt1p-treatment (unmodified U), and the unmodified tRNA presents a canonical core structure as defined in mol A but not mol B in the Up-modified structure. Do the authors mean even the tRNA is Up-modified, the tRNA can adopt different structures, and only the B conformation stabilizes the structure? (This might explain the small differences in Fig. 2a, although Fig. 2b convinces this reviewer that Up47 stabilizes tRNA *in vitro*.)

Response: As described in Discussion, U^P47 does not rigidly stabilize the tRNA, but flexibly deals with the structural change between Mol. A and B due to thermal denaturation of the core structure. As a metaphor, we liken U^P47 to a padlock. So, Mol. B is a metastable structure stabilized by U^P47. On the other hand, G⁺15 tightly fixes the core structure like a screw bolt by connecting the D- and V-loops to the acceptor stem.

Regarding the melting curves in Figure 1f, this is not a small change. Increased T_m value by 6.6°C is one of the largest values conferred by a single modification.

(2) The electron density for Up47 is very clear. However, could the authors provide Fo – Fc to better present the modification?

Response: We provided the simulated annealing omit Fo-Fc map for U^P47 in Mol. A (Extended Data Fig. 5c) and Mol. B (Extended Data Fig. 5d).

(3) (minor) Could the authors please change the color in Fig. 3e? It is hard (for this reviewer and maybe other readers) to distinguish the white and light cyan as indicated.

Response: As suggested, the color codes in Figure 3e,f have been changed.

4. It is exciting to see the authors provide solid bioinformatics, biochemistry, and structural evidence (1.8 Å resolution for ArkI) that ArkI and KptA are the kinase and the phosphatase.

(1) Could the authors comment on whether ArkI specifically modifies tRNA but no other RNAs? Some cursory analysis or comments on other natural RNA targets would have been helpful.

Response: According to our analyses, ArkI specifically introduces U^P47 in class I tRNAs bearing a V-loop with five bases. Class I tRNAs having four and six bases in the V-loop are not phosphorylated. In addition, class II tRNAs are not phosphorylated as well. These observations indicate strict substrate specificity of ArkI. To examine other RNAs, we carried out *in vitro* phosphorylation of total RNA with ArkI homologs in the presence of [γ -³²P] ATP (Supplementary Fig. 5b,c). After the reaction, total RNA was resolved by PAGE, and ³²P-labeled RNAs were visualized by fluorimager. As expected, TkArkI phosphorylated class I tRNA fraction, no other RNAs including class II tRNAs and rRNAs were phosphorylated. For other ArkI homologs, we examined *in vitro* phosphorylation of *E. coli* total RNA, and found that only class I tRNAs were phosphorylated. These results suggest that class I tRNAs are major substrates for ArkI homologs specifically. However, we cannot rule out a possibility that some other mRNAs and non-coding RNAs are phosphorylated by ArkI if they have tRNA-like structures.

(2) How does ArkI specifically recognize tRNA? Does ArkI harbor a structure-specific, tRNA-binding motif? This motif generally recognizes the outside corner of the L-shaped tRNA structure, which is the signature of a tRNA molecule. For example, as the Arc1p domain of

yeast MetRS, the EMAPII domain of human TyrRS, and the freestanding Trpb111 of the *A. aeolicus* thermophile. Some analysis of the sequence of ArkI that aligns with those orthologs would be helpful.

Response: Thank you very much for the interesting comment. We carefully looked for some motifs related to tRNA binding, but could not. ArkI belongs to a superfamily of eukaryotic protein kinases. The electrostatic surface potential reveals a large positive area in one side of the TkArkI structure. Mutation studies revealed that the basic surface in the C-terminal lobe might bind substrate tRNA through electrostatic interaction.

(3) Based on Fig. 4c, is ArkI the only tRNA kinase in this thermophilic species?

Response: Yes. ArkI is the only tRNA kinase responsible for U^P47 in this organism. But, we don't know yet whether there are other tRNA kinases for different positions.

5. The biochemistry evidence is clear that ATP is the phosphate donor. However, could the authors comment on:

(1) Whether tRNA binding increased ATP binding affinity (especially ATP cannot be seen in the structure of enzyme alone)?

Response: We solved a crystal structure of the guanosine-bound TkArkI. In this structure, the P-loop and other motifs responsible for ATP binding are dislocated from the guanosine binding site (Fig. 4c,d). So, we speculated that the P-loop and other motifs form the active pocket for ATP binding upon tRNA binding to TkArkI. We did not directly measure ATP binding affinity to TkArkI, but instead, we have measured kinetic parameters for ATP in U^P47 formation. Compared to submicro molar K_m for tRNA (0.097 μ M), K_m for ATP was quite high, it is 1.2 mM, indicating that TkArkI has a low affinity to ATP even in the presence of tRNA. Moreover, we here found that U^P47 formation is sensitive to cellular ATP concentration. We added relevant description and discussion on this additional finding.

(2) What the role of GTP identified in the structures is? Dose GTP have a higher binding affinity than ATP, so that GTP stays in the structure but not for catalysis? Is the GTP binding site the same as the ATP binding site (might be out of the scope of the study)?

Response: In our crystal structure of TkArkI, guanosine (not GTP) was tightly bound. As described in Discussion, we don't know biological meaning of the guanosine-binding to TkArkI. Guanosine may compete with ATP to regulate tRNA phosphorylation. In this revision, we determined K_m value for ATP (1.2 mM). This value is extremely high when compared to that of known protein kinases (they usually have micromolar range of K_m for ATP). If TkArkI senses cellular energy metabolism, guanosine-binding to TkArkI might have a regulatory role in U^P47 formation. However, we cannot rule out the possibility that guanosine is an artifact bound to the inactive form of TkArkI, because TkArkI is a recombinant protein expressed in *E. coli*. It is unclear whether guanosine actually binds to TkArkI within archaeal cells at high growth temperature.

6. The structure of ArkI is *T. kodakarensis* ArkI; could the authors comment on why they express ArkI of *N. viennensis* in *E. coli* in Fig. 6?

Response: Because *N. viennensis* is a mesophilic archaeon, we considered that *N. viennensis*

tokodaii, we don't have to consider reversibility of N324 modification in this organism.

5. On p. 7, rephrase the statement "Compared with the tRNA with Up 47, the hypomodified tRNA degraded within 5 min" more precisely.

Response: We rephrased it as follows, "Compared with the intact tRNA having U^p47, Tpt1p-treated tRNA degraded more rapidly within 5 min."

Reviewer Reports on the First Revision:

Referee #2:

The authors have addressed all my previous concerns!